# Fast Fractional Natural Gradient Descent using Learnable Spectral Factorizations

## Abstract

Many popular optimization methods can be united through fractional natural gradient descent (FNGD), which pre-conditions the gradient with a fractional power of the inverse Fisher: RMSprop and Adam(W) estimate a diagonal Fisher matrix and apply a square root before inversion; other methods like K-FAC and Shampoo employ matrix-valued Fisher estimates and apply the inverse and inverse square root, respectively. Recently, the question of how fractional power affects optimization has moved into focus, e.g. offering trade-offs between convergence and generalization. Gaining deeper insights into this phenomenon would require going beyond diagonal estimations and using cheap and flexible matrix-valued Fisher estimators capable of applying any fractional power; however, existing methods are limited by their expensive matrix fraction computation. To address this, we propose a Riemannian framework to learn eigen-factorized Fisher estimations on the fly, allowing for the cheap application of *arbitrary* fractional powers. Our approach does not require matrix decompositions and, therefore, is stable in half precision. We show our framework's efficacy on positive-definite matrix optimization problems and demonstrate its efficiency and flexibility for training neural nets.

## 1 Introduction

Many well-known adaptive methods, like SGD (Robbins & Monro, 1951), RmsProp (Tieleman & Hinton, 2012) or Adam(W) (Kingma & Ba, 2015; Loshchilov & Hutter, 2017), can be framed as fractional natural gradient descent (FNGD): Given the neural network (NN) parameters $\boldsymbol{\mu}$, the gradient $\mathbf{g}$ and a curvature estimation $\mathbf{S}$, FNGD applies $\boldsymbol{\mu} \leftarrow \boldsymbol{\mu} - \beta_1 \mathbf{S}^{-1/p} \mathbf{g}$ using a learning rate $\beta_1$ and subjecting the curvature approximation to a fractional power $1/p$ before inversion. $\mathbf{S}$ approximates the Fisher information matrix (Amari, 1998), e.g. through exponential averages of the empirical Fisher (Kunstner et al., 2019) or the gradient outer product (GOP, Kingma & Ba, 2015; Agarwal et al., 2019; Lin et al., 2024). FNGD's matrix fractional power allows interpolating between NGD (Amari, 1998) with $p = 1$ and SGD as $p \to \infty$; RMSprop/Adam(W) use $p = 2$.

While most adaptive optimization algorithms rely on a square root ($p = 2$), the fractional power's role has recently garnered a lot of attention. Several works question the indispensable role of the square root and empirically demonstrate the usage of other fractions to trade off convergence and generalization (Chen et al., 2021), overcome the generalization gap between SGD and adaptive methods on convolutional neural nets (CNNs) observed by (Wilson et al., 2017), and to successfully train transformers (Lin et al., 2024). Theoretically, Huh (2020) identify limitations of SGD and NGD in terms of generalization, convergence, and stability when training deep linear networks. They argue that FNGD with $p \notin \{1, \infty\}$ can offer the 'best of both worlds', i.e. NGD's convergence speed with SGD's generalization.

Applying other fractional powers is straightforward for methods with a diagonal curvature approximation (e.g. PAdam from Chen et al., 2021) and does not add much computational cost. However, doing so for methods with non-diagonal preconditioning matrices such as K-FAC (Martens & Grosse, 2015) or Shampoo (Gupta et al., 2018; Anil et al., 2020; Shi et al., 2023) is computationally and numerically challenging. This is because computing *matrix* fractional powers is computationally intensive, and must usually be done in high precision to avoid numerical instabilities (Anil et al., 2020; Shi et al., 2023), preventing those methods from using fast, low-precision arithmetic (Micikevicius et al., 2018). Making the root computation fast and stable in low-precision can further unleash the potential of non-diagonal fractional methods.

To catalyze further investigations into the fractional power's role, it would be desirable to have a flexible and efficient framework for learning non-diagonal curvature approximations that can (i) apply arbitrary fractional roots, and (ii) circumvent the numerical instabilities of matrix decompositions.

We address this instability and inefficiency and present an update scheme to directly adapt the spectral factorization $\mathbf{B}\text{diag}(\mathbf{d})\mathbf{B}^\top$ of the curvature approximation $\mathbf{S}$ on the fly, which we term a spectral parametrization of $\mathbf{S}$. Thanks to this factorization, we can apply any matrix fractional power to $\mathbf{S}$ by elementwise operation on the eigenvalues $\mathbf{d}$. Our approach directly adapts eigenfactors and maintains the factorization, and a practical version can operate without performing eigendecompositions. This makes our scheme amenable to running in low precision because we do not use any unstable matrix decomposition algorithm. However, the spectral factorization introduces several challenges as it imposes constraints and ambiguities (i.e. $\mathbf{B}$ must be orthogonal and $\mathbf{d}$ sorted) that need to be dealt with. These constraints make it more challenging to maintain the factorization on the fly. Our contributions are:

- We propose an update scheme to learn the spectral factorization $\mathbf{B}\text{diag}(\mathbf{d})\mathbf{B}^\top$ of a curvature matrix $\mathbf{S}$ on the fly and address how to account for the constraints, and resolve the ambiguities, imposed by this parameterization. We then show how to learn Kronecker-factorized spectral decompositions, i.e. $\mathbf{S} \approx (\mathbf{S}^{(K)} \otimes \mathbf{S}^{(C)})$ where $\mathbf{S}^{(i)} = \mathbf{B}^{(i)}\text{diag}(\mathbf{d}^{(i)})\mathbf{B}^{(i)^\top}$, which are crucial to scale the approach. The Kronecker factorization introduces new ambiguities, which we resolve by introducing a scalar $\alpha$ and demanding $\det(\mathbf{d}_i) = 1$ (Section 2).

- Similar to Glasmachers et al. (2010); Lin et al. (2021; 2023), our approach views learning the curvature approximation as learning the covariance of a Gaussian variational distribution by performing Riemannian gradient descent on the manifold of dense or Kronecker-factorized positive-definite matrices. We extend these works by incorporating the new constraints arising from the spectral decomposition for the Fisher-Rao metric. (Section 3.)

- Empirically, we demonstrate the effectiveness of our approach for a range of applications, including positive-definite matrix optimization (Pennec et al., 2006; Absil et al., 2009) and low-precision neural net training.

## 1.1 BACKGROUND

To train an NN model, we solve an unconstrained optimization problem. The objective function of the problem is expressed as a finite sum of cost functions with $N$ observations:

$$\min_\mu \ell(\boldsymbol{\mu}) := \sum_{i=1}^N c(f(\mathbf{x}_i; \boldsymbol{\mu}), y_i), \tag{1}$$

where $\mathbf{x}_i$ and $y_i$ are features and a label for the $i$-th observation, respectively, $f(\cdot; \boldsymbol{\mu})$ is an NN with learnable weights $\boldsymbol{\mu}$, and $c(\cdot, y_i)$ is a cost function such as the cross-entropy function to measure the difference between the output of the NN and label $y_i$.

We consider adaptive methods to solve this problem, where we estimate a preconditioning matrix by only using gradient information. For many well-known adaptive methods such as RMRprop (Tieleman & Hinton, 2012), a square root (i.e., $p = 2$) is introduced.

$$\text{RmsProp} : \mathbf{S} \leftarrow (1 - \beta_2)\mathbf{S} + \beta_2 \text{diag}(\mathbf{g}\mathbf{g}^T), \ \boldsymbol{\mu} \leftarrow \boldsymbol{\mu} - \beta_1 \mathbf{S}^{-1/p}\mathbf{g}, \tag{2}$$

where $\mathbf{S}$ is a diagonal matrix and $\mathbf{g} = \nabla_\mu \ell$ is a gradient vector of the objective function. We often estimate the vector using a mini-batch of observations. Lin et al. (2024) consider a full matrix version of the root-free RmsProp update scheme (i.e., $p = 1$) and propose an inverse-free update scheme. Other works improve the performance of adaptive methods on CNNs using other roots, such as $p = 4$ in Chen et al. (2021) and $p = 1$ in Lin et al. (2024).

$$\mathbf{S} \leftarrow (1 - \beta_2)\mathbf{S} + \beta_2 \mathcal{H} = \mathbf{S} + \beta_2(\mathcal{H} - \mathbf{S}), \ \boldsymbol{\mu} \leftarrow \boldsymbol{\mu} - \beta_2 \mathbf{S}^{-1/p}\mathbf{g}, \tag{3}$$

Non-diagonal adaptive methods, like Shampoo (Gupta et al., 2018), also include a fractional root (e.g., $p = 4$) in their update rule.

$$\text{Shampoo} : \mathbf{S}_C \leftarrow (1 - \beta_2)\mathbf{S}_C + \beta_2 \mathbf{G}\mathbf{G}^T, \ \mathbf{S}_K \leftarrow (1 - \beta_2)\mathbf{S}_K + \beta_2 \mathbf{G}^T\mathbf{G}, \tag{4}$$

$$\boldsymbol{\mu} \leftarrow \boldsymbol{\mu} - \beta_1 (\mathbf{S}_K \otimes \mathbf{S}_C)^{-1/p} \mathbf{g} \iff \mathbf{M} \leftarrow \mathbf{M} - \beta_1 \mathbf{S}_C^{-1/p} \mathbf{G} \mathbf{S}_K^{-1/p}, \tag{5}$$

where $\mathbf{M} = \text{Mat}(\boldsymbol{\mu})$ and $\mathbf{G} = \text{Mat}(\mathbf{g})$ are matrix representations of $\boldsymbol{\mu}$ and $\mathbf{g}$, respectively.

## 2 FAST FNGD USING LEARNABLE SPECTRAL FACTORIZATIONS

Our goal is to design pre-conditioner update schemes that offer the flexibility to apply arbitrary matrix roots at a low cost. The starting point is the observation that the pre-conditioner $\mathbf{S}$ can be interpreted as the inverse covariance matrix of a Gaussian variational distribution (Lin et al., 2024), which can be learned on the fly via the update scheme in Equation (3). However, this parameterization complicates applying any fractional root to $\mathbf{S}$ since the root computation requires matrix decomposition.

Our contribution is to propose an update scheme for a new parameterization $\mathbf{S} = \mathbf{B}\mathrm{diag}(\mathbf{d})\mathbf{B}^\top$, where $\mathbf{B}$ is an orthogonal square matrix and $\mathbf{d}$ is a vector with positive sorted entries, to learn $\mathbf{d}$ and $\mathbf{B}$. We call this parameterization a *spectral parameterization*, due to its connection to the spectral decomposition of symmetric matrices. We empirically and theoretically establish its equivalence to update rules that directly update $\mathbf{S}$, implying one can enjoy efficient updates while retaining the behavior of traditional methods. Our spectral parametrization allows us to easily compute any fractional root $\mathbf{S}^{-1/p} = \mathbf{B}\mathrm{diag}(\mathbf{d}^{-1/p})\mathbf{B}^\top$ through elementwise roots on $\mathbf{d}$, instead of matrix roots on $\mathbf{S}$. We then extend it to Kronecker-factorized matrices, which is crucial for large-scale applications. Our update schemes are efficient as learning $\mathbf{B}$ and $\mathbf{d}$ does not involve any matrix decomposition.

In contrast to previous parameterizations, the spectral parameterization introduces new challenges, such as satisfying the orthogonal constraint on $\mathbf{B}$ and handling parametrization ambiguities. We defer the technicalities how to handle these constraints to Section 3 where we derive the update scheme from scratch. This is mainly to avoid introducing technical Riemannian optimization concepts needed that are necessary for our derivation. In the following, our focus will be on presenting and empirically validating the update scheme, and its connections to existing methods.

### 2.1 FULL-MATRIX ADAPTIVE SCHEMES THROUGH A FULL GAUSSIAN APPROXIMATION

We first present our update scheme in the context of full-matrix preconditioners. While full-matrix preconditioners are generally impractical for modern neural networks, this will serve to illustrate the core ideas which we later apply to structured preconditioners. We obtain the update scheme with $p = 1$ by solving a Gaussian problem with mean $\boldsymbol{\mu}$ and reparametrized inverse covariance $\mathbf{S} = \mathbf{B}\mathrm{Diag}(\mathbf{d})\mathbf{B}^T$. We will discuss the procedure to obtain the scheme and satisfy the spectral constraints in Sec. 3.3. Given a learnable spectral parametrization, our update scheme shown in the leftmost box of Fig. 1 allows us to introduce any fractional $p$-root further and efficiently compute the root when updating $\boldsymbol{\mu}$.

We theoretically establish the equivalence of this update to the default scheme as stated in:

*Claim* 1. Our update scheme in the leftmost box of Fig 1 is equivalent to the scheme in Equation (3) up to first-order accuracy in $\beta_2$ when $\mathbf{d}$ does not have repeated entries (proof in Appendix C).

**Empirical validation of the Full-matrix Update Scheme**  We empirically evaluate our scheme on $\mathbf{S} = \mathbf{B}\mathrm{Diag}(\mathbf{d})\mathbf{B}^T$ for curvature approximation. We compare our scheme to the default training scheme on $\mathbf{S}$ as $\mathbf{S}_{k+1} \leftarrow (1 - \beta)\mathbf{S}_k + \beta \mathbf{g}_k \mathbf{g}_k^T$, and the inverse-free scheme (Lin et al., 2024) for a learnable Cholesky factorization $\mathbf{C}$ of $\mathbf{S}^{-1}$ (i.e., $\mathbf{S} = (\mathbf{C}\mathbf{C})^{-1}$). We focus on the preconditioner estimation of $\mathbf{S}$ based on a fixed gradient sequence $\{\mathbf{g}_1, \ldots, \mathbf{g}_T\}$ and initialized by the same $\mathbf{S}_0$. We consider two scenarios in this evaluation: (1) fixed-point matching and (2) iterate matching.

**Fixed-point matching**  The ground truth in this setting is a fixed-point solution, $\mathbf{S}_* = E[\mathbf{g}\mathbf{g}^T] = \boldsymbol{\Sigma}$, to the default update scheme as $\mathbf{S}_* = (1 - \beta)\mathbf{S}_* + \beta \mathbf{g}_k \mathbf{g}_k^T$, where $\mathbf{g}_k$ is independently generated from a normal distribution $\mathbf{g}_k \sim \mathcal{N}(\mathbf{0}, \boldsymbol{\Sigma})$ at each iteration $k$. We evaluate an update scheme in every iteration $k$ by comparing its current estimate denoted by $\mathbf{S}_k^{(est)}$ to the fixed point. We use a relative Frobenius norm $\frac{\|\mathbf{S}_* - \mathbf{S}_k^{(est)}\|_F}{\|\mathbf{S}_*\|_F}$ and the Wasserstein-2 distance for positive-definite matrices to measure the difference.

**Iterate matching**  The ground truth is a sequence of matrices $\{\mathbf{S}_1^{(true)}, \ldots, \mathbf{S}_T^{(true)}\}$ generated by the default scheme when applying the scheme to the gradient sequence. We are interested in matching the iterate that the default scheme generates at every step. We use a relative Frobenius norm $\frac{\|\mathbf{S}_k^{(true)} - \mathbf{S}_k^{(est)}\|_F}{\|\mathbf{S}_k^{(true)}\|_F}$ and the Wasserstein-2 distance to measure the discrepancy between an update scheme and the default update scheme at every iteration $k$.

**Full-matrix** $(\mathbf{S} = \mathrm{BDiag}(\mathbf{d})\mathbf{B}^T)$

1: Compute gradient $\mathbf{g} := \nabla\ell(\boldsymbol{\mu})$

$\quad \mathbf{d} \leftarrow \mathbf{d} \odot \exp\{\beta_2 \mathbf{d}^{-1} \odot [-\mathbf{d} + \mathrm{diag}(\mathbf{B}^T\mathbf{g}\mathbf{g}^T\mathbf{B})]\}$

$\quad \mathbf{B} \leftarrow \mathbf{B}\mathrm{Cayley}(\beta_2(\mathrm{Skew}(\mathrm{Tril}(\mathbf{U}))))$

2: $\boldsymbol{\mu} \leftarrow \boldsymbol{\mu} - \beta_1 \mathbf{B}\mathrm{Diag}(\mathbf{d}^{-1/p})\mathbf{B}^T\mathbf{g}$

**Kronecker** $\big(\mathbf{s} = \alpha[\mathbf{B}^{(C)}\mathrm{Diag}(\mathbf{d}^{(C)})(\mathbf{B}^{(C)})^T] \otimes [\mathbf{B}^{(K)}\mathrm{Diag}(\mathbf{d}^{(K)})(\mathbf{B}^{(K)})^T]\big)$

1: Compute gradient $\mathbf{G} := \mathrm{Mat}(\nabla\ell(\boldsymbol{\mu}))$

$\quad \mathbf{m}^{(C)} = (\mathbf{d}^{(C)})^{-1} \odot [-\mathbf{d}^{(C)} + \frac{1}{\alpha m}\mathrm{diag}(\mathbf{W}^{(C)})]$

$\quad \mathbf{m}^{(K)} = (\mathbf{d}^{(K)})^{-1} \odot [-\mathbf{d}^{(K)} + \frac{1}{\alpha n}\mathrm{diag}(\mathbf{W}^{(K)})]$

$\quad \mathbf{d}^{(C)} \leftarrow \mathbf{d}^{(C)} \odot \exp\{\beta_2[\mathbf{m}^{(C)} - \mathrm{mean}(\mathbf{m}^{(C)})]\}$

$\quad \mathbf{d}^{(K)} \leftarrow \mathbf{d}^{(K)} \odot \exp\{\beta_2[\mathbf{m}^{(K)} - \mathrm{mean}(\mathbf{m}^{(K)})]\}$

$\quad \mathbf{B}^{(C)} \leftarrow \mathbf{B}^{(C)}\mathrm{Cayley}(\frac{\beta_2}{\alpha m}\mathrm{Skew}(\mathrm{Tril}(\mathbf{U}^{(C)})))$

$\quad \mathbf{B}^{(K)} \leftarrow \mathbf{B}^{(K)}\mathrm{Cayley}(\frac{\beta_2}{\alpha n}\mathrm{Skew}(\mathrm{Tril}(\mathbf{U}^{(K)})))$

$\quad \alpha \leftarrow \alpha\exp(\frac{\beta_2}{2}[\mathrm{mean}(\mathbf{m}^{(K)}) + \mathrm{mean}(\mathbf{m}^{(C)})])$

2: $\mathbf{M} \leftarrow \mathbf{M} - \beta_1(\alpha^{-1/p})(\mathbf{S}^{(C)})^{-1/p}\mathbf{G}(\mathbf{S}^{(K)})^{-1/p}$

Figure 1: Adaptive update schemes for full-matrix and Kronecker structured spectral factorization for a finite sum of loss functions. Both update schemes use map $\mathrm{Cayley}(\mathbf{N}) := (\mathbf{I} + \mathbf{N})(\mathbf{I} - \mathbf{N})^{-1}$ with skew-symmetric $\mathbf{N} = -\mathbf{N}^\top$ to output an orthogonal matrix, map $\mathrm{Skew}(\mathbf{M}) := \mathbf{M} - \mathbf{M}^\top$ to skew-symmetrize an arbitrary square matrix, and map $\mathrm{Tril}(\cdot)$ returns a lower-triangular matrix with zero diagonal entries. $\odot$ denotes the elementwise product. For simplicity, we assume NN weights take a matrix form: $\mathbf{M} := \mathrm{Mat}(\boldsymbol{\mu}) \in \mathbb{R}^{n \times m}$. **Full-matrix scheme:** matrix $\mathrm{Tril}(\mathbf{U})$ is a lower-triangular matrix (i.e., $i > j$) with the $(i,j)$-th entry $[U]_{ij} := -[\mathbf{B}^T\mathbf{g}\mathbf{g}^T\mathbf{B}]_{ij}/(d_i - d_j)$ when $d_i \neq d_j$ and 0 otherwise. **Kronecker-based scheme:** This update scheme uses $\mathbf{W}^{(C)} := (\mathbf{B}^{(C)})^T\mathbf{G}(\mathbf{S}^{(K)})^{-1}\mathbf{G}^T\mathbf{B}^{(C)}$, and $\mathbf{W}^{(K)} := (\mathbf{B}^{(K)})^T\mathbf{G}^T(\mathbf{S}^{(C)})^{-1}\mathbf{G}\mathbf{B}^{(K)}$, where $\mathbf{S}^{(K)} := \mathbf{B}^{(K)}\mathrm{Diag}(\mathbf{d}^{(K)})(\mathbf{B}^{(K)})^T \in \mathbb{R}^{m \times m}$ and $\mathbf{S}^{(C)} := \mathbf{B}^{(C)}\mathrm{Diag}(\mathbf{d}^{(C)})(\mathbf{B}^{(C)})^T \in \mathbb{R}^{n \times n}$ is easy to compute due to the spectral factorization. For each Kronecker factor $\mathbf{S}^{(C)}$, matrix $\mathrm{Tril}(\mathbf{U}^{(C)})$ is a lower-triangular matrix with its $(i,j)$-th entry $[U^{(C)}]_{ij} := -[W^{(C)}]_{ij}/([d^{(C)}]_i - [d^{(C)}]_j)$ if $[d^{(C)}]_i \neq [d^{(C)}]_j$ and 0 otherwise, where $[d^{(C)}]_i$ denotes the $i$-th entry of vector $\mathbf{d}^{(C)}$. Vector $\mathbf{d}^{(C)}$ satisfies the determinant constraint $\det(\mathrm{diag}(\mathbf{d}^{(C)})) = 1$ since $\mathrm{sum}(\mathbf{m}^{(C)} - \mathrm{mean}(\mathbf{m}^{(C)})) = 0$ (Sec. 3.2). For low-precision NN training, we truncate the Cayley and the exponential map.

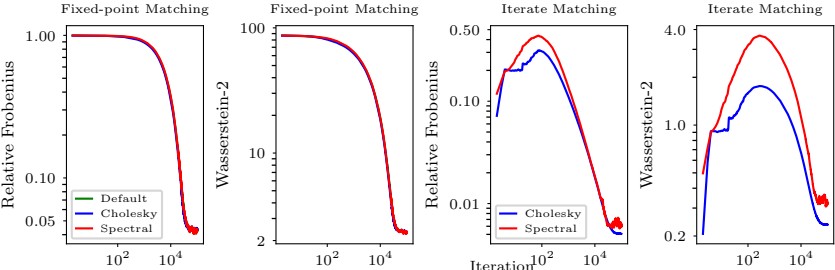

Figure 2: Empirical validation of our full-matrix update scheme on estimating a preconditioner $\mathbf{S} \in \mathbb{R}^{100 \times 100}$. The first two figures on the left show that our update scheme converges to a fixed-point solution as fast as the default update scheme in $\mathbf{S}$ and the Cholesky-based scheme. The last two figures illustrate how closely our update scheme matches the iterates generated by the default update scheme at each iteration. Our update scheme and the Cholesky-based scheme perform similarly for matching the preconditioner estimates generated by the default scheme.

From Fig. 2, we can see that our update scheme performs similarly to the default update scheme in the two scenarios. These results demonstrate the empirical equivalence between our scheme and the default scheme, at least for curvature estimation.

## 2.2 Kronecker-structured Schemes through a matrix Gaussian Approximation

Using Kronecker-structured preconditioners (Martens & Grosse, 2015; Gupta et al., 2018) is necessary for large models as a full-matrix preconditioner is too large to store. Many Kronecker-based methods (Zhang et al., 2018; Ren & Goldfarb, 2021; Lin et al., 2023; 2024) are based on a (matrix) Gaussian family with Kronecker-structured inverse covariance $\mathbf{S} = \mathbf{S}^{(C)} \otimes \mathbf{S}^{(K)}$. However, many Kronecker-based methods depend on a particular choice of Kronecker factorization because Kronecker factorization is not unique. As will be discussed in Sec. 3.2, we make the factorization unique by imposing a determinant constraint on each factor and introducing a learnable scalar $\alpha$. Consequently, we consider this Kronecker structure $\mathbf{S} = \alpha[\mathbf{S}^{(C)} \otimes \mathbf{S}^{(K)}]$ with constraints $\det(\mathbf{S}^{(C)}) = \det(\mathbf{S}^{(K)}) = 1$ and $\alpha > 0$. We then propose a spectral parametrization for each Kronecker factor while satisfying these constraints. The update scheme is presented in Fig.1.

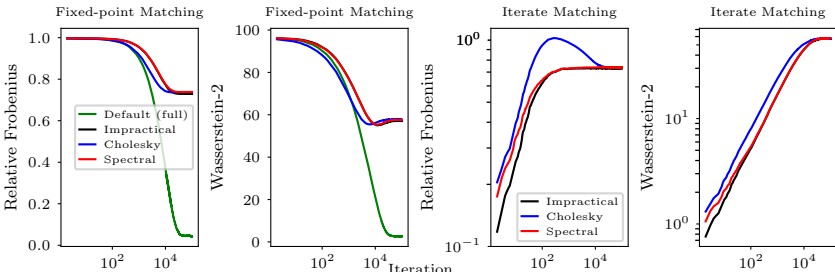

Figure 3: Empirical validation of our Kronecker-structured update scheme on estimating a precondi-tioner $\mathbf{S} \approx \mathbf{S}^{(C)} \otimes \mathbf{S}^{(K)}$, where $\mathbf{S}^{(C)} \in \mathbb{R}^{9 \times 9}$ and $\mathbf{S}^{(K)} \in \mathbb{R}^{11 \times 11}$. The first two figures on the left show that our update scheme gives a structural approximation of a fixed-point solution that obtained by the default full-matrix update scheme. Our scheme converges as fast as Kronecker-structured baseline methods, including the impractical projection-based method. The last two figures illustrate how closely our scheme matches the unstructured iterates generated by the default scheme at each iteration. All update schemes perform similarly due to the structural approximation gap.

**Empirical Evaluation of the Kronecker-based Update Scheme**    Similarly, we evaluate our struc-tured scheme for curvature approximation. Our goal is to obtain a Kronecker-structured estimation of $\mathbf{S}$. We compare our scheme to the default unstructured scheme on $\mathbf{S}$: $\mathbf{S}_{k+1} \leftarrow (1 - \beta)\mathbf{S}_k + \beta \mathbf{g}_k \mathbf{g}_k^T$. As baselines, we consider the curvature estimation used in the structured Cholesky factorization (Lin et al., 2024), and an impractical projection-based method (Van Loan & Pitsianis, 1993): $(\mathbf{S}_{k+1}^{(C)}, \mathbf{S}_{k+1}^{(K)}) \leftarrow \text{Proj}((1 - \gamma)(\mathbf{S}_k^{(C)} \otimes \mathbf{S}_k^{(K)}) + \gamma \mathbf{g}_k \mathbf{g}_k^T)$. We use a similar experimental setup and consider two similar scenarios discussed in Sec. 2.1. Here, we initialized all update schemes by a Kronecker structured matrix $\mathbf{S}_0$ to remove the difference introduced by initialization:

**Fixed-point matching**  The ground truth is an unstructured fixed-point solution, $\mathbf{S}_* = E[\mathbf{g}\mathbf{g}^T] = \mathbf{\Sigma}$. We evaluate a Kronecker-structured scheme in every iteration $k$ by comparing its current structured estimate to the fixed point. We measure the difference using the same metrics considered previously.

**Iterate matching**  The ground truth is a sequence of unstructured matrices generated by the default scheme. Our goal is to match the iterate that the default scheme generates using Kronecker structured approximations We use the same metrics to measure the difference.

From Fig. 3, we can see that our structural scheme performs as well as structural baselines. Our approach even performs similarly to the impractical method that requires storing a full matrix and solving a projection optimization problem at every iteration. This illustrates the effectiveness of our approach in Kronecker-structured cases.

## 2.3 CONNECTIONS TO DIAGONAL METHODS

Our update scheme in Fig. 1 also applies in diagonal cases by forcing $\mathbf{B}$ to be a diagonal matrix. We achieve that by changing map $\text{Tril}(\cdot)$ to $\text{Diag}(\cdot)$ in the update rule. Consequently, $\mathbf{B}$ becomes an identity matrix up to sign changes. Similar to the full matrix case, we can obtain this scheme through a diagonal Gaussian approximation. When truncating the exponential map, our scheme becomes the root-free RMSprop (Lin et al., 2024). If applying a fractional $p$-root, our scheme also recovers RMSprop (Tieleman & Hinton, 2012) for $p = 2$ and the fractional diagonal method (Chen et al., 2021) for $p = 4$. See Appx. B for the detail.

## 2.4 NUMERICAL APPROXIMATIONS FOR COST REDUCTION AND LOW-PRECISION TRAINING

Our update scheme can be slow because the Cayley map involves computationally expensive matrix inversion that requires high-precision floating-point arithmetic to avoid numerical instability. Like other works (Liu et al., 2021; Li et al., 2020; Qiu et al., 2023), we consider a truncated Cayley map for NN problems to work with lower precision and reduce cost while maintaining numerical stability. Our truncation is based on a Neumann series for the matrix inversion (Krishnan et al., 2017; Lorraine et al., 2020; Qiu et al., 2023). This is possible because we can approximate the matrix inversion in the Cay-ley map $\text{Cayley}(\beta\mathbf{N}) = (\mathbf{I} + \beta\mathbf{N})(\mathbf{I} - \beta\mathbf{N})^{-1} = (\mathbf{I} + \beta\mathbf{N}) \prod_{l=0}^{\infty} (\mathbf{I} + (\beta\mathbf{N})^{2^l}) \approx (\mathbf{I} + \beta\mathbf{N})^2 (\mathbf{I} + (\beta\mathbf{N})^2)(\mathbf{I} + (\beta\mathbf{N})^4)$

based on a convergent Neumann series, when $\beta$ is small enough so that $\|\beta\mathbf{N}\| < 1$, where $\beta := 1 - \hat{\beta}_2$ and $\hat{\beta}_2$ is Adam's $\beta_2$, see Appx. B for the detail.

## 3 LEARNING SPECTRAL FACTORIZATIONS VIA COORDINATE TRANSFORMS

Here, we derive our update schemes for learning a spectral factorization on the fly. Our starting point is that, according to Lin et al. (2024), a root-free method in (3) is a Riemannian solution to a Gaussian approximation problem in a particular coordinate. Because Riemannian methods are invariant under coordinate transformations, our idea is to change coordinates so that the Riemannian solution becomes a root-free update rule for spectral factorization in new coordinates.

**Riemannian Approach for Obtaining Root-free Update Schemes** Lin et al. (2024) show that a root-free adaptive update scheme is a simplified version of Riemannian gradient descent (RGD) (c.f., Eq. (7) ) on a Gaussian manifold (Amari, 2016), where $\boldsymbol{\mu}$ and $\mathbf{S}$ in the root-free scheme become Gaussian's mean and inverse covariance, respectively. They consider a Gaussian approximation problem and use the following procedure to obtain the adaptive update scheme in (3) with $p = 1$,

**Step 1** They first reformulate the original problem in (1) as a Gaussian approximation:

$$\min_{\mu, S \succ 0} \mathcal{L}(\boldsymbol{\mu}, \mathbf{S}) := E_{w \sim q(w; \mu, S)}[\ell(\mathbf{w})] - \mathcal{Q}_q, \tag{6}$$

where $\ell(\cdot)$ is the loss function in the original problem, a new symbol $\mathbf{w}$ is used to denote the weights of the NN because they are no longer learnable, $q(\mathbf{w}; \boldsymbol{\mu}, \mathbf{S})$ is a Gaussian with mean $\boldsymbol{\mu}$ and covariance $\mathbf{S}^{-1}$, and $\mathcal{Q}_q := E_{w \sim q}[-\log q(\mathbf{w}; \boldsymbol{\mu}, \mathbf{S})] = -\frac{1}{2}\log\det(\mathbf{S})$ is the Gaussian's differential entropy.

**Step 2** They then suggest performing RGD in a parameter space $\boldsymbol{\tau} := \{\boldsymbol{\mu}, \mathbf{S}\}$ of the Gaussian.

$$\text{RGD} : \boldsymbol{\tau} \leftarrow \boldsymbol{\tau} - \beta[\mathbf{F}_\tau]^{-1}\nabla_\tau\mathcal{L}, \tag{7}$$

where $\mathbf{F}_\tau := -E_{w \sim q}[\nabla_\tau^2 \log q(\mathbf{w}; \boldsymbol{\tau})] \in \mathbb{R}^{(l+l^2) \times (l+l^2)}$ is the Fisher-Rao metric representation in $\boldsymbol{\tau}$ and $l$ is the number of NN weights. The metric is also known as the affine-invariant metric (Pennec et al., 2006) for positive-definite matrices (Minh & Murino, 2017) when the mean $\boldsymbol{\mu}$ is constant.

**Step 3** Simplifying the RGD step

$$\text{RGD} : \begin{bmatrix} \boldsymbol{\mu} \\ \mathbf{S} \end{bmatrix} \leftarrow \begin{bmatrix} \boldsymbol{\mu} \\ \mathbf{S} \end{bmatrix} - \beta \begin{bmatrix} \mathbf{S}^{-1} & \mathbf{0} \\ \mathbf{0} & -2\frac{\partial\mathbf{S}}{\partial\mathbf{S}^{-1}} \end{bmatrix} \begin{bmatrix} \partial_\mu\mathcal{L} \\ \partial_S\mathcal{L} \end{bmatrix} = \begin{bmatrix} \boldsymbol{\mu} - \beta\mathbf{S}^{-1}\partial_\mu\mathcal{L} \\ \mathbf{S} + \beta(2\partial_{S^{-1}}\mathcal{L}) \end{bmatrix} \approx \begin{bmatrix} \boldsymbol{\mu} - \beta\mathbf{S}^{-1}\mathbf{g} \\ \mathbf{S} + \beta(\mathcal{H} - \mathbf{S}) \end{bmatrix}, \tag{8}$$

gives rise to the root-free adaptive scheme in (3), where they use (i) the analytical inverse metric $[\mathbf{F}_\tau]^{-1} = \begin{bmatrix} \mathbf{S}^{-1} & \mathbf{0} \\ \mathbf{0} & -2\frac{\partial\mathbf{S}}{\partial\mathbf{S}^{-1}}, \end{bmatrix}$, (ii) Stein's identities for the Gaussian (Opper & Archambeau, 2009), and (iii) a valid Hessian approximation (Lin et al., 2024) $\mathcal{H} = \mathbf{g}\mathbf{g}^T$ with a delta evaluation at the mean $\boldsymbol{\mu}$:

$$\partial_\mu\mathcal{L} \stackrel{\text{Stein}}{=} E_{w \sim q}[\nabla_w\ell] \stackrel{\text{delta}}{\approx} \nabla_\mu\ell = \mathbf{g}, \quad 2\partial_{S^{-1}}\mathcal{L} \stackrel{\text{Stein}}{=} E_{w \sim q}[\nabla_w^2\ell] - \mathbf{S} \stackrel{\text{delta}}{\approx} \nabla_\mu^2\ell - \mathbf{S} \approx \mathcal{H} - \mathbf{S}. \tag{9}$$

**Challenges of Learning Spectral Parametrizations via RGD** Our main idea is to learn a spectral parameterization/coordinate of $\mathbf{S} = \mathbf{B}\text{Diag}(\mathbf{d})\mathbf{B}^T$ by solving a reparametrized Gaussian problem in Eq. 6. We then follow a similar procedure to convert an RGD step in the spectral coordinate into a root-free update scheme based on the spectral factorization. However, directly performing RGD in a spectral coordinate is challenging because we have to (1) satisfy parameter constraints, (2) use a non-singular metric (coordinate representation), and (3) analytically compute the metric inversion.

**Conflict between Simplification for RGD and Coordinate Transformation** The simplification step (Step 3) turns a computationally expensive RGD step involving the metric inversion into a more efficient root-free adaptive update scheme. Without an analytical metric inversion, we cannot simplify the RGD step and explicitly express it as a root-free update scheme. When changing coordinates, the metric representation has to be changed accordingly (Lee, 2018) to make RGD invariant to coordinate transformation. However, the coordinate change of the metric complicates the simplification. This is because we no longer *analytically* inverse this high-dimensional Fisher-Rao metric, which is non-diagonal and singular in some coordinates like a spectral coordinate. The simplification process is more challenging in Kronecker-factorized cases because additional redundancy from Kronecker factorization renders the metric (coordinate representation) singular and complicates the simplification.

**Constraint Satisfaction and Metric Diagonalization via *Local* Coordinate Transformation**
Inspired by general Riemannian (normal) local coordinates (Glasmachers et al., 2010; Lin et al., 2021; 2023) and Fermi coordinates (Manasse & Misner, 1963), we propose using local coordinates to tackle these challenges. The main idea is to construct local coordinates that handle constraints and facilitate the analytical metric inversion needed for the simplification. Using a local coordinate transformation can *locally* diagonalize the metric at a *single* evaluation point. Given a global coordinate, such as a spectral coordinate, we construct a local coordinate and its coordinate transformation map associated with the global coordinate at every iteration. In our approach, the coordinate and its transformation map should satisfy these conditions: (1) the map is differentiable and injective. (2) the coordinate should not have any coordinate constraint, and its origin represents a current iterate in the spectral coordinate. (3) it is common for the metric evaluated at the origin of the local coordinate to be an identity matrix (Lin et al., 2023). However, this can be loosened to be diagonal (Glasmachers et al., 2010) or even block-diagonal as long as the metric is easy to inverse. Our approach follows the requirements for local coordinate construction (Lin et al., 2023). We propose new local coordinates for spectral coordinates because the existing local coordinates do not account for spectral constraints.

### 3.1 Handling Spectral Parameter Constraints and Redundancies

Here, we describe our local coordinates and coordinate transformation maps for handling constraints for spectral factorization. Through a coordinate transformation map, we express the spectral constraints using an unconstrained local coordinate. Because a spectral coordinate contains redundancies, we remove them to simplify the inverse metric computation. This allows us to make the metric diagonal in our local coordinates.

**Handling Constraints via *Local* Coordinate Transformation** A spectral coordinate has parameter constraints: $\mathbf{B}$ is an orthogonal square matrix, and $\mathbf{d}$ is a vector with positive entries. We consider Cayley[1] and exponential maps to construct transformation maps, where we introduce a map denoted by $\mathrm{Skew}(\cdot)$ to make its input skew-symmetric as required by the Cayley map. At every iteration $k$, we handle the constraints (c.f., Claim (2)) by constructing a local parametrization in $(\mathbf{m}, \mathbf{M})$ associated to the current iteration $(\mathbf{d}_k, \mathbf{B}_k)$ through a local transformation map :

$$( \mathbf{d}(\mathbf{m}), \mathbf{B}(\mathbf{M}) ) = ( \mathbf{d}_k \odot \exp(\mathbf{m}), \mathbf{B}_k \mathrm{Cayley}(\mathrm{Skew}(\mathrm{Tril}(\mathbf{M}))) ), \qquad (10)$$

where $\odot$ is the elementwise product, $(\mathbf{d}_k, \mathbf{B}_k)$ represents a point evaluated at the $k$-th iteration. This transformation map is *locally* defined because it depends on a current point $(\mathbf{d}_k, \mathbf{B}_k)$ that changes at every iteration. We use the origin in this local system to represent $(\mathbf{d}_k, \mathbf{B}_k)$. This is possible because $(\mathbf{d}(\mathbf{m}), \mathbf{B}(\mathbf{M}))\big|_{m=0, M=0}$ becomes $(\mathbf{d}_k, \mathbf{B}_k) = (\mathbf{d}(\mathbf{0}), \mathbf{B}(\mathbf{0}))$ when evaluating this map (10) at the origin. The origin simplifies the metric computation as many terms arising in the metric vanish.

*Claim* 2. The map in Eq. (10) is differentiable and injective. We satisfy the parameter constraints in a spectral coordinate by changing a local coordinate to the spectral coordinate through the map.

**Removing Redundancy due to Spectral Factorization** A spectral parametrization contains redundancy due to permutation. For simplicity, we assume all entries of $\mathbf{d}$ are distinct in the following discussion and will later address cases when $\mathbf{d}$ has repeated values. For example, consider another eigen factorization: $\mathbf{S} = \bar{\mathbf{B}}\mathrm{Diag}(\bar{\mathbf{d}})\bar{\mathbf{B}}^T$, where $\bar{\mathbf{B}} := \mathbf{B}\mathbf{Q}$ and $\bar{\mathbf{d}}$ are permuted values so that $\mathrm{Diag}(\bar{\mathbf{d}}) = \mathbf{Q}^T\mathrm{Diag}(\mathbf{d})\mathbf{Q}$. To remove this redundancy, we restrict $\mathbf{M}$ to be a *lower-triangular* matrix explicitly denoted by $\mathrm{Tril}(\mathbf{M})$. In eigendecomposition, $\mathbf{d}$ (as a vector of eigenvalues) is ranked to disallow any permutation. We consider an alternative solution by restricting $\mathbf{B}$ because we can not simultaneously rank and learn $\mathbf{d}$ on the fly. Restricting $\mathbf{B}$ to remove this redundancy means $\mathbf{B}(\mathbf{M}_1) = \mathbf{B}(\mathbf{M}_2)\mathbf{Q}$ holds only when $\mathbf{M}_1 \equiv \mathbf{M}_2$ in the local coordinate. The lower-triangular restriction of $\mathbf{M}$ removes the redundancy because $\mathbf{M}_1 \equiv \mathbf{M}_2$ as shown in Claim 3. Here, we assume the lower-triangular restriction, $\mathrm{Tril}(\mathbf{M})$, also makes the diagonal entries of its input, $\mathbf{M}$, zero. Otherwise, $\mathbf{M}_1$ and $\mathbf{M}_2$ can differ in their diagonal entries because the skew-symmetrization in the transformation map (Eq. (10)) always ignores these entries.

*Claim* 3. Given that entries of $\mathbf{d}$ are not duplicated, a spectral parameterization obtained through the map in (10) is unique under permutations when using a lower-triangular restriction.

---

[1]We use the Cayley map to construct $\mathbf{B}$, where $\mathbf{B}$ is special orthogonal (i.e., $\det(\mathbf{B}) = 1$). Although the map does not represent all special orthogonal matrices, it is widely used in practice (Li et al., 2020; Liu et al., 2021).

**Handling Redundancy due to Repeated Entries of $\mathbf{d}$** Recall that eigendecomposition is not unique when having repeat eigenvalues. Our spectral parametrization is also not unique when $\mathbf{d}$ has duplicated entries. In this case, the Fisher-Rao metric (coordinate representation) is singular. We allow $\mathbf{d}$ to have repeated entries and address the singularity using the Moore–Penrose inversion (van Oostrum et al., 2023). Computing the Moore–Penrose inversion is easy because we diagonalize the metric at evaluation points. In practice, we also use this inversion to improve numerical stability if $\mathbf{d}$ has very close entries.

## 3.2 Handling Constraints and Redundancies for Kronecker Structures

Now, we propose spectral parametrizations and local coordinates for Kronecker structured matrices. Kronecker factorization introduces an additional redundancy that makes the exact Fisher-Rao metric singular and non-block-diagonal in Kronecker structured coordinate. We construct local coordinates to remove this redundancy and simplify the inverse metric computation. Furthermore, removing this redundancy makes our update scheme invariant to all equivalent Kronecker factorizations.

**Removing Redundancy due to Kronecker Factorization** Using a Kronecker structure introduces redundancy because Kronecker factorization is non-unique. For example, we can reexpress a structured matrix $\mathbf{S} = \mathbf{S}^{(C)} \otimes \mathbf{S}^{(K)}$ in another way: $\mathbf{S} = \gamma \otimes [\bar{\mathbf{S}}^{(C)} \otimes \bar{\mathbf{S}}^{(K)}]$, where $\bar{\mathbf{S}}^{(C)} := \gamma^{-1/2} \mathbf{S}^{(C)}$, $\bar{\mathbf{S}}^{(K)} := \gamma^{-1/2} \mathbf{S}^{(K)}$, and $\gamma > 0$ is a learnable scalar. Without removing this redundancy, learning the factorization $(\mathbf{S}^{(C)}, \mathbf{S}^{(K)})$ is not equivalent to learning another factorization $(\gamma, \bar{\mathbf{S}}^{(C)}, \bar{\mathbf{S}}^{(K)})$ (i.e., $\mathbf{S} = \mathbf{S}^{(C)} \otimes \mathbf{S}^{(K)} \neq \bar{\mathbf{S}} = \gamma \otimes \bar{\mathbf{S}}^{(C)} \otimes \bar{\mathbf{S}}^{(K)}$ ). We eliminate this redundancy by imposing a determinant constraint on each Kronecker factor and adding an extra scalar $\alpha$. This leads to a unique representation: $\mathbf{S} := \alpha [\mathbf{S}^{(C)} \otimes \mathbf{S}^{(K)}] = \bar{\alpha} [\gamma \otimes \bar{\mathbf{S}}^{(C)} \otimes \bar{\mathbf{S}}^{(K)}]$ because the constraint $\det(\gamma) = 1$ makes the scalar $\gamma$ constant, where $\det(\mathbf{S}^{(C)}) = \det(\mathbf{S}^{(K)}) = \det(\bar{\mathbf{S}}^{(C)}) = \det(\bar{\mathbf{S}}^{(K)}) = \det(\gamma) = 1$, $\alpha > 0$, and $\bar{\alpha} > 0$. Consequently, we propose a spectral parametrization for each Kronecker factor $\mathbf{S}^{(C)} = \mathbf{B}^{(C)} \mathrm{Diag}(\mathbf{d}^{(C)})(\mathbf{B}^{(C)})^T$ with $\det(\mathrm{Diag}(\mathbf{d}^{(C)})) = 1$ to satisfy the determinant constraint.

**Handling Constraints via *Local* Coordinate Transformation** We construct local coordinates to handle constraints, including the determinant constraints. For the positive scalar $\alpha$, we introduce a local coordinate $n$ and use an exponential map in the coordinate transformation: $\alpha(n) = \alpha_k \exp(n)$ at every iteration $k$. For each Kronecker factor, we drop the factor index for simplicity and construct a local coordinate $(\mathbf{m}, \mathbf{M})$. The coordinate transformation map for each factor

$$\mathbf{d}(\mathbf{m}) = \mathbf{d}_k \odot \exp(\mathbf{m}), \quad \mathbf{B}(\mathbf{M}) = \mathbf{B}_k \mathrm{Cayley}(\mathrm{Skew}(\mathrm{Tril}(\mathbf{M}))), \tag{11}$$

is similar to the map (c.f., Eq. 10) in full-matrix cases, expect that we require $\mathrm{sum}(\mathbf{m}) = 0$ to satisfy the determinant constraint (i.e., $\det(\mathrm{Diag}(\mathbf{d}(\mathbf{m}))) = 1$), where $l$ is the length of vector $\mathbf{d}_k$ and the local coordinate $\mathbf{m} := [m_1, \ldots, m_{l-1}, -\sum_i^{l-1} m_i]$ has only $(l-1)$ free variables.

## 3.3 Derivation of Root-free Update Schemes through Gaussian Approximations

Now, we present a procedure to obtain root-free update schemes for our spectral parametrizations. Our procedure follows similar steps suggested by Lin et al. (2024) except that we use local coordinates. A similar procedure can solve positive-definite matrix optimization problems (Pennec et al., 2006).

**Procedure for Full-matrix Spectral Parametrizations** To derive full-matrix root-free update schemes, we follow these three steps.

**Step 1** We solve a Gaussian problem similar to (6) using a learnable spectral parametrization, $\mathbf{S} = \mathbf{B} \mathrm{Diag}(\mathbf{d}) \mathbf{B}^T$, of the inverse covariance:

$$\min_{\boldsymbol{\mu}, d, B} \mathcal{L}(\boldsymbol{\mu}, \mathbf{d}, \mathbf{B}) := E_{w \sim q(w; \mu, d, B)}[\ell(\mathbf{w})] - \mathcal{Q}_{q(\mu, d, B)}, \quad \text{s.t.} \quad \mathbf{BB}^T = \mathbf{B}^T \mathbf{B} = \mathbf{I}, \ \mathbf{d} > 0. \tag{12}$$

**Step 2** We construct a local coordinate at each iteration to remove the (spectral) constraints in (12) when performing RGD. We then take an RGD step without constraints in this local coordinate and translate the change from the local coordinate to the spectral coordinate.

**Step 2.1** Concretely, at iteration $k$, we create a local coordinate $\boldsymbol{\eta} := (\boldsymbol{\delta}, \mathbf{m}, \mathbf{M})$ at the current point $\boldsymbol{\tau}_k := (\boldsymbol{\mu}_k, \mathbf{d}_k, \mathbf{B}_k)$ and use this local transformation map

$$\boldsymbol{\tau}(\boldsymbol{\eta}; \boldsymbol{\tau}_k) := \begin{bmatrix} \boldsymbol{\mu}(\boldsymbol{\delta}; \boldsymbol{\tau}_k) \\ \mathbf{d}(\mathbf{m}; \boldsymbol{\tau}_k) \\ \mathbf{B}(\mathbf{M}; \boldsymbol{\tau}_k) \end{bmatrix} = \begin{bmatrix} \boldsymbol{\mu}_k + \mathbf{B}_k \mathrm{Diag}(\mathbf{d}_k^{-1/2}) \boldsymbol{\delta} \\ \mathbf{d}_k \odot \exp(\mathbf{m}) \\ \mathbf{B}_k \mathrm{Cayley}(\mathrm{Skew}(\mathrm{Tril}(\mathbf{M}))) \end{bmatrix}, \tag{13}$$

to translate the change from the local coordinate to the spectral coordinate (c.f., Claim 2), where $\boldsymbol{\tau}_k = (\boldsymbol{\mu}_k, \mathbf{d}_k, \mathbf{B}_k)$ is considered as a constant in this map, and $\mathrm{Tril}(\mathbf{M})$ is used to explicitly enforce the lower-triangular restriction (c.f., Sec. 3.1).

**Step 2.2** We then take an (unconstrained) RGD step in the local coordinate,

$$\mathrm{RGD}: \boldsymbol{\eta}_{\mathrm{new}} \leftarrow \boldsymbol{\eta}_{\mathrm{cur}} - \beta[\mathbf{F}_\eta(\boldsymbol{\eta}_{\mathrm{cur}})]^{-1}\nabla_\eta\mathcal{L}\big|_{\eta:=\eta_{\mathrm{cur}}} = \mathbf{0} - \beta[\mathbf{F}_\eta(\mathbf{0})]^{-1}\nabla_\eta\mathcal{L}\big|_{\eta:=0}, \quad (14)$$

and translate the change $\boldsymbol{\eta}_{\mathrm{new}}$ from the local coordinate

$$\boldsymbol{\tau}_{k+1} \leftarrow \boldsymbol{\tau}(\boldsymbol{\eta}_{\mathrm{new}}; \boldsymbol{\tau}_k), \quad (15)$$

to the eigen coordinate, where the Fisher-Rao metric $\mathbf{F}_\eta(\boldsymbol{\eta}_{\mathrm{cur}})$ evaluated at $\boldsymbol{\eta}_{\mathrm{cur}}$ is diagonal (c.f., Claim 4) in the local coordinate and the origin $\boldsymbol{\eta}_{\mathrm{cur}} \equiv \mathbf{0}$ represents the current point $\boldsymbol{\tau}_k = \boldsymbol{\tau}(\boldsymbol{\eta}_{\mathrm{cur}}; \boldsymbol{\tau}_k) \equiv \boldsymbol{\tau}(\mathbf{0}; \boldsymbol{\tau}_k)$ in the spectral coordinate.

**Step 3** We obtain the root-free update scheme in Fig. 1 by simplifying this RGD step in (14)-(15), and making the same approximations in (9). The simplification is easy because the metric $\mathbf{F}_\eta(\boldsymbol{\eta})\big|_{\eta=\eta_{\mathrm{cur}}}$ evaluated at the origin in the local coordinate is diagonal. This allows us to simplify the inverse metric computation in Eq. (14) even when the metric is singular. Moreover, the gradient $\nabla_\eta\mathcal{L}\big|_{\eta=\eta_{\mathrm{cur}}}$ required by RGD is easy to compute via the chain rule and has an analytical expression. See Appx. H for a complete derivation.

*Claim* 4. **Metric Diagonalization:** The exact Fisher-Rao metric $\mathbf{F}_\eta(\boldsymbol{\eta}_{\mathrm{cur}})$ (for a full Gaussian) evaluated at the origin $\boldsymbol{\eta}_{\mathrm{cur}} \equiv \mathbf{0}$ is diagonal and has a closed-form expression:

$$\mathbf{F}_\eta(\boldsymbol{\delta}, \mathbf{m}, \mathrm{vecTril}(\mathbf{M}))\big|_{\boldsymbol{\delta}=0, \mathbf{m}=0, \mathbf{M}=0} = \begin{bmatrix} \mathbf{F}_{\delta\delta} & \mathbf{0} & \mathbf{0} \\ \mathbf{0} & \mathbf{F}_{mm} & \mathbf{0} \\ \mathbf{0} & \mathbf{0} & \mathbf{F}_{MM} \end{bmatrix} \quad (16)$$

where $\mathbf{F}_{\delta\delta} = \mathbf{I}$, $\mathbf{F}_{mm} = \frac{1}{2}\mathbf{I}$, $\mathbf{F}_{MM} = \mathrm{Diag}(\mathrm{vecTril}(\mathbf{C}))$, $\mathrm{vecTril}(\mathbf{C})$ represents the low-triangular half of $\mathbf{C}$ excluding diagonal entries and its $(i, j)$-th entry is $[C]_{ij} = 4(\frac{d_i}{d_j} + \frac{d_j}{d_i} - 2) \geq 0$ and $d_i$ denotes the $i$-th entry of $\mathbf{d}_k$. The metric is singular when $\mathbf{d}$ has repeated entries (i.e., $d_i = d_j$ for $i \neq j$). Consequently, we can use the Moore-Penrose inversion to inverse the metric (c.f., Sec. 3.1).

**Discussion about the Induced Metric for Orthogonal Matrix B** Our approach implicitly constructs a Riemannian metric for the orthogonal matrix $\mathbf{B}$ through the coordinate transformation of the Fisher-Rao metric of a Gaussian. This induced metric for the orthogonal matrix differs from existing metrics (Tagare, 2011; Li et al., 2020; Kong et al., 2022) in the Riemannian optimization literature. We use the Fisher-Rao metric because our goal is to learn an orthogonal matrix for spectral factorization.

**Procedure for Kronecker-structured Spectral Parametrizations** We use a similar procedure to obtain our update schemes in a structured case. We briefly describe the procedure and highlight the difference in this case.

**Step 1** We solve a Gaussian problem similar to (6) using a Kronecker factorized eigenparametrization, $\mathbf{S} = \alpha[\mathbf{B}^{(C)}\mathrm{Diag}(\mathbf{d}^{(C)})(\mathbf{B}^{(C)})^T] \otimes [\mathbf{B}^{(K)}\mathrm{Diag}(\mathbf{d}^{(K)})(\mathbf{B}^{(K)})^T]$, for the inverse covariance with extra constraints: $\det(\mathrm{Diag}(\mathbf{d}^{(C)})) = \det(\mathrm{Diag}(\mathbf{d}^{(K)})) = 1$ and $\alpha > 0$.

**Step 2** We construct a local coordinate at each iteration, perform RGD in the local coordinate, and translate the change using a transformation map in Eq. (11). See Eq. (35) in the appendix for details.

**Step 3** We obtain the root-free update scheme in Fig. 1 by simplifying this RGD step. The simplification is straightforward because the metric evaluated at the origin in the local coordinate is block diagonal (c.f., Claim 5).

*Claim* 5. **Metric Block Diagonalization:** The exact Fisher-Rao metric $\mathbf{F}_\eta(\boldsymbol{\eta}_{\mathrm{cur}})$ (for a matrix Gaussian) evaluated at the origin $\boldsymbol{\eta}_{\mathrm{cur}} \equiv \mathbf{0}$ is block-diagonal and has a closed-form expression. The inverse metric also has an analytical form.

## 4 EXPERIMENTS

We first consider a positive-definite matrix optimization problem to validate our full-matrix update scheme beyond NN training. We aim to learn a positive-definite matrix $\mathbf{S}$ from noisy observations that can be negative-definite. Therefore, a linear update scheme to update $\mathbf{S}$ like the one in Eq. (3) is unsuitable for the problem because the scheme assumes an observation (e.g., a gradient outer product) is

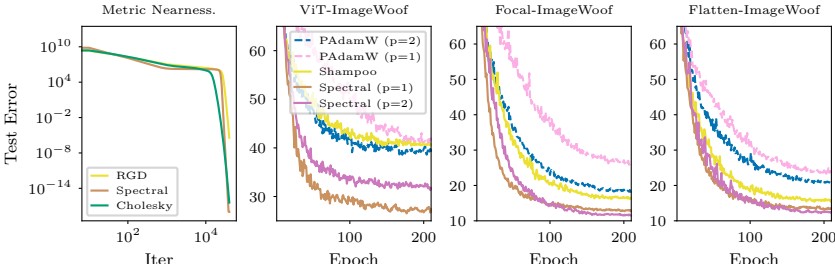

Figure 4: Experiments demonstrate effectiveness and efficiency of our update schemes. The first figure on the left shows the performance of our full-matrix update scheme for learning positive-definite matrices. Our update scheme matches the baselines as our scheme is RGD in local coordinates. The remaining three figures show the performance of our Kronecker update scheme for training vision transformers with low precision. To match AdamW's running time (i.e, PAdamW with $p = 2$), we can update our perconditioner at every 10 iterations because we use a truncated Cayley map. Shampoo has to update its perconditioner at every 100 iterations to match the running time. This is because Shampoo performs eigendecompision when updating its preconditioner. We use grafting to improve Shampoo's performance. Without grafting, Shampoo barely outperforms AdamW and is hard to tune due to the infrequent update of its preconditioner.

semi-positive-definite. Given that our approach is RGD in local coordinates, we consider the standard RGD with retraction (Absil et al., 2009) and the Cholesky-based RGD (Lin et al., 2023) as baselines. We consider the metric nearness problem (Brickell et al., 2008) $\min_{W \succ 0} \ell(\mathbf{W}) := \frac{1}{2N} \sum_{i=1}^{N} \|\mathbf{W}\mathbf{Q}\mathbf{x}_i - \mathbf{x}_i\|_2^2$ used in the matrix optimization literature , where $\mathbf{Q} \in \mathbb{R}^{d \times d}$ is a known positive-definite matrix and only a subset of $\mathbf{x}_i \in \mathbb{R}^d$ are observed at each iteration. The ground truth is $\mathbf{W}_* = \mathbf{Q}^{-1}$ and we measure the difference between an estimate $\mathbf{W}_{\text{est}}$ and the ground truth $\mathbf{W}_*$ using $\ell(\mathbf{W}_{\text{est}}) - \ell(\mathbf{W}_*)$. We consider a case for $d = 60$ and generate $\mathbf{Q}$ and $\mathbf{x}_i$. As we can see from the leftmost plot in Fig. 4, our method performs as well as the RGD-based methods. This result shows the potential of our scheme for positive-definite matrix optimization beyond NN training.

Next, we examine our Kronecker-structured update scheme in low-precision NN training problems. We use our update scheme to train vision transformers from scratch with half-precision. Training transformers in half-precision allows us to evaluate the numerical stability of our approach because matrix methods g can be unstable in low precision. We then show the effectiveness and efficiency of our approach by comparing our method to strong baselines like AdamW (i.e., PAdamW with $p = 2$) and Shampoo. We consider training three vision transformers: ViT (Dosovitskiy, 2020), FocalNet (Yang et al., 2022), and FlattenViT (Han et al., 2023), on the ImageWoof dataset using mini-batches with batch size 128. Our method updates its preconditioner at every 10 iterations to match AdmaW's runtime because it does not require matrix decomposition and inversion when using a truncated Cayley map. Shampoo has to update its preconditioner at every 100 iterations to match AdmaW's runtime because of eigendecomposition. This also shows the low iteration cost of our method as our preconditioner can be updated more frequently. We use the state of the art implementation of Shampoo (Shi et al., 2023). We have to use grafting (Agarwal et al., 2021) to improve Shampoo's performance due to the infrequency update of the preconditioner. We use random search (Choi et al., 2019) to tune all available hyperparameters for each method using 200 runs. From the remaining plots in Fig. 4, we can see that our method effectively trains transformers with low-precision and often outperforms these baselines. Moreover, our method is flexible enough to use other fractional roots. From the second plot on the left in Fig. 4, we can see that other roots such as $p = 1$ is better than the square root $p = 2$. This shows that the potential of using other fractional roots.

## 5 CONCLUSION

We present a Riemannian approach for learning spectral factorizations on the fly. Our method fixes the instability and inefficiency of using a matrix fractional root for low-precision NN training and enables matrix methods to use other fractional roots. An interesting direction is to evaluate our methods in large-scale settings and investigate the potential benefit of using other fractional roots.

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

## A    PROPERTIES OF THE CAYLEY MAP

*Claim* 6. The Cayley map $\mathrm{Cayley}(\mathbf{N}) = (\mathbf{I} + \mathbf{N})(\mathbf{I} - \mathbf{N})^{-1}$ is well-defined for skew-symmetric $\mathbf{N}$. Moreover, this map is injective.

*Proof.* To show the map is well-defined, we want to show $(\mathbf{I} - \mathbf{N})$ is non-singular. Suppose not, we have $\det(\mathbf{I} - \mathbf{N}) = 0$. Thus, $\mathbf{N}$ has an eigenvalue with 1. By the definition of the eigenvalue, there exists a non-zero vector $\mathbf{x} \neq \mathbf{0}$ so that $\mathbf{N}\mathbf{x} = \mathbf{x}$. Notice that Given that $\mathbf{N}$ is skew-symmetric, we have

$$\mathbf{N} + \mathbf{N}^T = \mathbf{0} \tag{17}$$

and

$$0 = \mathbf{x}^T(\mathbf{N} + \mathbf{N}^T)\mathbf{x} = \mathbf{x}^T(\mathbf{N}\mathbf{x}) + (\mathbf{x}^T\mathbf{N}^T)\mathbf{x} = \mathbf{x}^T\mathbf{x} + \mathbf{x}^T\mathbf{x} \tag{18}$$

The above expression implies $\mathbf{x} = \mathbf{0}$, which is a contradiction. Thus, $\det(\mathbf{I} - \mathbf{N}) \neq 0$ and $(\mathbf{I} - \mathbf{N})$ is non-singular.

Let $\mathbf{Q} = \mathrm{Cayley}(\mathbf{N})$. We show that the Cayley is injective if $\mathbf{N}$ is skew-symmetric. We first assume $(\mathbf{Q} + \mathbf{I})$ is non-singular and then we prove it. Given $(\mathbf{Q} + \mathbf{I})$ is non-singular, we have

$$\mathbf{Q}(\mathbf{I} - \mathbf{N}) = (\mathbf{I} + \mathbf{N}) \iff \mathbf{Q} - \mathbf{I} = (\mathbf{Q} + \mathbf{I})\mathbf{N} \iff (\mathbf{Q} + \mathbf{I})^{-1}(\mathbf{Q} - \mathbf{I}) = \mathbf{N},$$

This implies the map is injective and its inverse is

$$\mathbf{N} = \mathrm{Cayley}^{-1}(\mathbf{Q}) := (\mathbf{Q} + \mathbf{I})^{-1}(\mathbf{Q} - \mathbf{I}) \tag{19}$$

Now, we show that $(\mathbf{Q} + \mathbf{I})$ is non-singular. We use proof by contradiction. If not, there exists a non-zero vector $\mathbf{v}$ (Maddocks, 2021) so that

$$\mathbf{Q}\mathbf{v} = -\mathbf{v} \iff (\mathbf{I} + \mathbf{N})(\mathbf{I} - \mathbf{N})^{-1}\mathbf{v} = -\mathbf{v} \tag{20}$$
$$\iff (\mathbf{I} - \mathbf{N})^{-1}(\mathbf{I} + \mathbf{N})\mathbf{v} = -\mathbf{v} \tag{21}$$
$$\iff (\mathbf{I} + \mathbf{N})\mathbf{v} = -(\mathbf{I} - \mathbf{N})\mathbf{v} \tag{22}$$
$$\iff \mathbf{v} = -\mathbf{v}, \text{ (another contradiction since } \mathbf{v} \neq 0) \tag{23}$$

where we use the following identity in the second step in the above expression.

$$(\mathbf{I} + \mathbf{N})(\mathbf{I} - \mathbf{N})^{-1} = -(-2\mathbf{I} + \mathbf{I} - \mathbf{N})(\mathbf{I} - \mathbf{N})^{-1} = 2(\mathbf{I} - \mathbf{N})^{-1} - \mathbf{I} \tag{24}$$
$$= (\mathbf{I} - \mathbf{N})^{-1}(2\mathbf{I} - (\mathbf{I} - \mathbf{N})) = (\mathbf{I} - \mathbf{N})^{-1}(\mathbf{I} + \mathbf{N}) \tag{25}$$

$\square$

## B    CONNECTION TO DIAGONAL METHODS

Here, we show the connections between our scheme and the RmsProp method. Observe that eigenvalues are diagonal entries of a diagonal preconditioning matrix (i.e., $\mathbf{d} = \mathrm{diag}(\mathbf{S})$). Because $\mathbf{B}$ is now a diagonal and orthogonal matrix, each diagonal entry can only be 1 or -1. Using this result, we can further simplify our update scheme and recover the RmsProp update rules when using a first-order truncation of the exponential map.

$$\mathbf{d} \leftarrow \mathbf{d} \odot \exp\{\beta_2 \, \mathbf{d}^{-1} \odot [-\mathbf{d} + \mathrm{diag}(\mathbf{B}^T\mathbf{g}\mathbf{g}^T\mathbf{B})]\} \approx \mathbf{d} + \beta_2[-\mathbf{d} + \mathrm{diag}(\mathbf{g}\mathbf{g}^T)]\}$$

$$\mathbf{B} \leftarrow \mathbf{B}\mathrm{Cayley}(\frac{\beta_2}{2}\,\mathrm{Skew}(\mathrm{Diag}(\mathbf{U}))) = \mathbf{B}\mathrm{Cayley}(\mathbf{0}) = \mathbf{B}$$

$$\boldsymbol{\mu} \leftarrow \boldsymbol{\mu} - \beta_1\,\mathbf{B}\mathrm{Diag}(\mathbf{d}^{-1/p})\mathbf{B}^T\mathbf{g} = \boldsymbol{\mu} - \beta_1\,\mathrm{Diag}(\mathbf{d}^{-1/p})\mathbf{g}, \tag{26}$$

where $\mathrm{diag}(\mathbf{B}^T\mathbf{g}\mathbf{g}^T\mathbf{B}) = \mathrm{diag}(\mathbf{g}\mathbf{g}^T)$, $\mathbf{B}\mathrm{Diag}(\mathbf{d}^{-1/p})\mathbf{B}^T = \mathrm{Diag}(\mathbf{d}^{-1/p})$, and we use a first-order truncation of the exponential map $\mathbf{d} \odot \exp(\mathbf{d}^{-1} \odot \mathbf{n}) \approx \mathbf{d} \odot (\mathbf{1} + \mathbf{d}^{-1} \odot \mathbf{n}) = \mathbf{d} + \mathbf{n}$. Due to the skew-symmetrization, $\mathrm{Skew}(\mathrm{Diag}(\mathbf{U}))$ is always a zero matrix. Thus, according to our update scheme, $\mathbf{B}$ remains unchanged in diagonal cases because $\mathrm{Cayley}(\mathbf{0}) = \mathbf{I}$.

Our connections rely on a valid Hessian approximation (Lin et al., 2024) $\mathcal{H} = \mathbf{g}\mathbf{g}^T$. This requires the loss function in Eq. (1) can be expressed as a *normalized* negative log-likelihood such as the cross entropy loss. For example, an averaged version of the loss is not normalized, and therefore, the gradient outer product of the averaged loss is not a valid Hessian approximation.

# C    PROOF OF LEMMA 1

We will show that our update scheme in the leftmost box of Fig.1 is equivalent to the root-free update scheme in (3) up to first-order accuracy given that they use the same gradient $\mathbf{g}$. The proof can be easily generalized to every iteration by induction, given that both update schemes use the same sequence of gradients.

Recall that we update the spectral factors using the following rule at iteration $k$.

$$\mathbf{d}_{k+1} \leftarrow \mathbf{d}_k \odot \exp\{\beta_2 \, \mathbf{d}_k^{-1} \odot [-\mathbf{d}_k + \mathrm{diag}(\mathbf{B}_k^T \mathcal{H} \mathbf{B}_k)]\}$$

$$\mathbf{B}_{k+1} \leftarrow \mathbf{B}_k \mathrm{Cayley}(\frac{\beta_2}{2} \, \mathrm{Skew}(\mathrm{Tril}(\mathbf{U}))), \tag{27}$$

where $\mathcal{H} = \mathbf{g}\mathbf{g}^T$ and the $(i, j)$-th entry of $\mathbf{U}$ is $[U]_{ij} := -[\mathbf{B}_k^T \mathcal{H} \mathbf{B}_k]_{ij}/(d_i - d_j)$, where $\mathbf{d}$ has no repeated entries by our assumption. We want to show that the above update scheme is equivalent to the default update scheme up to first-order accuracy.

$$\mathbf{S}_{k+1} \leftarrow (1 - \beta_2)\mathbf{S}_k + \beta_2 \mathbf{g}_k \mathbf{g}_k^T \tag{28}$$

Let $\mathbf{Q}_k := \mathbf{B}_k \mathcal{H} \mathbf{B}_k^T$. Recall that the Cayley map is defined as $\mathrm{Cayley}(\mathbf{N}) = (\mathbf{I} + \mathbf{N})(\mathbf{I} - \mathbf{N})^{-1}$. Using the first-order approximation of $(\mathbf{I} - \mathbf{N})^{-1}$, we have $(\mathbf{I} - \beta_2\mathbf{N})^{-1} = \mathbf{I} + \beta_2\mathbf{N} + O(\beta_2^2)$. Thus, we have $\mathbf{B}_{k+1} = \mathbf{B}_k\mathrm{Cayley}(\frac{\beta_2}{2} \, \mathrm{Skew}(\mathrm{Tril}(\mathbf{U})) = \mathbf{B}_k(\mathbf{I} + \frac{\beta_2}{2}\mathbf{N})(\mathbf{I} + \frac{\beta_2}{2}\mathbf{N} + O(\beta_2^2)) = \mathbf{B}_k(\mathbf{I} + \beta_2\mathbf{N} + O(\beta_2^2))$, where $\mathbf{N} := \mathrm{Skew}(\mathrm{Tril}(\mathbf{U}))$. Similarly, we have $\mathbf{d}_{k+1} = \mathbf{d}_k \odot [1 + \beta_2\mathbf{d}_k^{-1} \odot (-\mathbf{d}_k + \mathrm{diag}(\mathbf{Q}_k)) + O(\beta_2^2)] = \mathbf{d}_k + \beta_2\mathbf{w}_k + O(\beta_2^2)$ by using the first-order truncation of the exponential map, where $\mathbf{w}_k := -\mathbf{d}_k + \mathrm{diag}(\mathbf{Q}_k)$. Notice that

$$\begin{aligned}
&\bar{\mathbf{S}}_{k+1} \\
=&\mathbf{B}_{k+1}\mathrm{Diag}(\mathbf{d}_{k+1})\mathbf{B}_{k+1}^T \\
=&\mathbf{B}_k\Big[(\mathbf{I} + \beta_2\mathbf{N} + O(\beta_2^2))\mathrm{Diag}(\mathbf{d}_k + \beta_2\mathbf{w}_k + O(\beta_2^2))\Big](\mathbf{I} + \beta_2\mathbf{N} + O(\beta_2^2))^T\mathbf{B}_k^T \\
=&\mathbf{B}_k\Big[\mathbf{D}_k + \beta_2\mathbf{N}\mathbf{D}_k + \beta_2\mathbf{W}_k + O(\beta_2^2)\Big](\mathbf{I} + \beta_2\mathbf{N} + O(\beta_2^2))^T\mathbf{B}_k^T \\
=&\mathbf{B}_k\Big[\mathbf{D}_k + \beta_2\mathbf{N}\mathbf{D}_k + \beta_2\mathbf{W}_k + \beta_2\mathbf{D}_k\mathbf{N}^T + O(\beta_2^2)\Big]\mathbf{B}_k^T \\
=&\mathbf{B}_k\mathbf{D}_k\mathbf{B}_k^T + \beta_2\mathbf{B}_k(\mathbf{N}\mathbf{D}_k + \mathbf{W}_k + \mathbf{D}_k\mathbf{N}^T)\mathbf{B}_k^T + O(\beta_2^2) \\
=&\bar{\mathbf{S}}_k + \beta_2\mathbf{B}_k(\mathbf{N}\mathbf{D}_k + \mathbf{D}_k\mathbf{N}^T)\mathbf{B}_k^T + \beta_2\mathbf{B}_k\mathbf{W}_k\mathbf{B}_k^T + O(\beta_2^2), \text{ (} \mathbf{N} \text{ is skew-symmetric)} \\
=&\bar{\mathbf{S}}_k + \beta_2\mathbf{B}_k(\mathbf{N}\mathbf{D}_k - \mathbf{D}_k\mathbf{N})\mathbf{B}_k^T + \beta_2\mathbf{B}_k\mathbf{W}_k\mathbf{B}_k^T + O(\beta_2^2)
\end{aligned}$$

where $\mathbf{D}_k := \mathrm{Diag}(\mathbf{d}_k)$ and $\mathbf{W}_k := \mathrm{Diag}(\mathbf{w}_k)$

Observation (1): Since $\mathbf{W}_k = \mathrm{Diag}(-\mathbf{d}_k + \mathrm{diag}(\mathbf{B}_k^T \mathcal{H} \mathbf{B}_k)) = -\mathbf{D}_k + \mathrm{Diag}(\mathrm{diag}(\mathbf{B}_k^T \mathcal{H} \mathbf{B}_k^T))$, we have

$$\mathbf{B}_k\mathbf{W}_k\mathbf{B}_k^T = -\mathbf{B}_k\mathbf{D}_k\mathbf{B}_k^T + \mathbf{B}_k\mathrm{Ddiag}(\mathbf{Q}_k)\mathbf{B}_k^T \tag{29}$$

where $\mathrm{Ddiag}(\mathbf{Q}_k)$ denotes the diagonal part of $\mathbf{Q}_k = \mathbf{B}_k\mathcal{H}\mathbf{B}_k^T$

Observation (2): Since $\mathbf{N} = \mathrm{Skew}(\mathrm{Tril}(\mathbf{U}))$ and $[\mathbf{U}]_{ij} = -[\mathbf{B}_k^T \mathcal{H} \mathbf{B}_k]_{ij}/(d_i - d_j) = -[\mathbf{Q}_k]_{ij}/(d_i - d_j)$, we can show that $\mathbf{N}\mathbf{D}_k - \mathbf{D}_k\mathbf{N}$ is indeed a symmetric matrix with zero-diagonal entries. Moreover, the low-triangular half $(i > j)$ of the matrix is

$$[\mathbf{N}\mathbf{D}_k - \mathbf{D}_k\mathbf{N}]_{ij} = (d_j - d_i)[\mathbf{U}]_{ij} = [\mathbf{Q}_k]_{ij}. \tag{30}$$

where $d_j \neq d_i$ since $\mathbf{d}$ has no repeated entries. Thus, we have $\mathbf{N}\mathbf{D}_k - \mathbf{D}_k\mathbf{N} = \mathbf{Q}_k - \mathrm{Ddiag}(\mathbf{Q}_k)$.

Using Observations (1) and (2), we have

$$\bar{\mathbf{S}}_{k+1}$$

$$=\mathbf{B}_k\Big[\mathbf{D}_k + \beta_2\mathbf{N}\mathbf{D}_k + \beta_2\mathbf{W}_k + \beta_2\mathbf{D}_k\mathbf{N}^T + O(\beta_2^2)\Big]\mathbf{B}_k^T$$

$$=\mathbf{B}_k\mathbf{D}_k\mathbf{B}_k^T + \beta_2\Big[\mathbf{B}_k\Big(\mathbf{Q}_k - \mathrm{Ddiag}(\mathbf{Q}_k)\Big)\mathbf{B}_k^T - \mathbf{B}_k\mathbf{D}_k\mathbf{B}_k^T + \mathbf{B}_k\mathrm{Ddiag}(\mathbf{Q}_k)\mathbf{B}_k^T\Big] + O(\beta_2^2)$$

$$=(1 - \beta_2)\mathbf{B}_k\mathbf{D}_k\mathbf{B}_k^T + \beta_2\mathbf{B}_k\Big[\mathbf{Q}_k\Big]\mathbf{B}_k^T + O(\beta_2^2), \text{ (Note: } \mathbf{Q}_k = \mathbf{B}_k^T\mathcal{H}\mathbf{B}_k)$$

$$=(1 - \beta_2)\bar{\mathbf{S}}_k + \beta_2\mathcal{H} + O(\beta_2^2),$$

which is exactly the default update scheme in (3) when dropping the second-order term $O(\beta_2^2)$.

## D    PROOF OF LEMMA 2

It is easy to see that the map is differentiable. We now show the map is injective. We only need to show the $\mathrm{Cayley}(\mathrm{Skew}(\mathrm{Tril}(\mathbf{M})))$ is injective. Since we only consider matrix $\mathbf{M}$ to have zero diagonal entries, it is equivalent to showing that the Cayley map is injective, which is true due to Claim 6.

Recall that the current point $(\mathbf{d}_k, \mathbf{B}_k)$ is in the spectral coordinate. Thus, we have $\mathbf{d}_k > 0$ and $\mathbf{B}_k$ is orthogonal. According to the map (10), it is easy to see that $\mathbf{d}(\mathbf{m}) = \mathbf{d}_k \odot \exp(\mathbf{m}) > 0$ because $\mathbf{d}_k > 0$. Thus, $\mathbf{d}(\mathbf{m})$ satisfies the parameter constraints. Now, we show that $\mathbf{B}(\mathbf{M})$ is also orthogonal. Because $\mathbf{B}_k$ is orthogonal, we only need to show that the output of the Cayley map, $\mathrm{Cayley}(\mathrm{Skew}(\mathrm{Tril}(\mathbf{M})))$, is orthogonal. Let $\mathbf{N} := \mathrm{Skew}(\mathrm{Tril}(\mathbf{M}))$. We know that $\mathbf{N}$ is skew-symmetric. We can verify that the Cayley transform satisfies the orthogonal constraint. Consider the following expression:

$$\big(\mathrm{Cayley}(\mathbf{N})\big)^T\mathrm{Cayley}(\mathbf{N}) = (\mathbf{I} - \mathbf{N})^{-T}(\mathbf{I} + \mathbf{N})^T(\mathbf{I} + \mathbf{N})(\mathbf{I} - \mathbf{N})^{-1} \tag{31}$$

$$= (\mathbf{I} - \mathbf{N})^{-T}(\mathbf{I} - \mathbf{N})(\mathbf{I} + \mathbf{N})(\mathbf{I} - \mathbf{N})^{-1} \tag{32}$$

$$= (\mathbf{I} - \mathbf{N})^{-T}(\mathbf{I} + \mathbf{N})(\mathbf{I} - \mathbf{N})(\mathbf{I} - \mathbf{N})^{-1} \tag{33}$$

$$= (\mathbf{I} - \mathbf{N})^{-T}(\mathbf{I} - \mathbf{N})^T(\mathbf{I} - \mathbf{N})(\mathbf{I} - \mathbf{N})^{-1} = \mathbf{I} \tag{34}$$

where we use the fact that $\mathbf{N}$ is skew-symmetric such as $\mathbf{N}^T = -\mathbf{N}$.

Likewise, we can show $\mathrm{Cayley}(\mathbf{N})\big(\mathrm{Cayley}(\mathbf{N})\big)^T = \mathbf{I}$. Thus, the output of the Cayley map is a square orthogonal matrix.

## E    PROOF OF LEMMA 3

Given the transformation map defined in Eq. (10), we want to show that $\mathbf{B}(\mathbf{M}_1) = \mathbf{B}(\mathbf{M}_2)\mathbf{Q}$ holds only when $\mathbf{M}_1 = \mathbf{M}_2$ for a permutation matrix $\mathbf{Q}$. By our definition, $\mathbf{M}_1$ and $\mathbf{M}_2$ must have zero diagonal entries.

Recall that $\mathbf{B}(\mathbf{M}) = \mathbf{B}_k\mathrm{Cayley}(\mathrm{Skew}(\mathrm{Tril}(\mathbf{M})))$ and $\mathrm{Cayley}(\mathbf{N}) = (\mathbf{I} + \mathbf{N})(\mathbf{I} - \mathbf{N})^{-1}$, where $\mathbf{N}$ is skew-symmetric. It is equivalent to show that this expression $\mathrm{Cayley}(\mathrm{Skew}(\mathrm{Tril}(\mathbf{M}_1))) = \mathrm{Cayley}(\mathrm{Skew}(\mathrm{Tril}(\mathbf{M}_2)))\mathbf{Q}$ holds only for $\mathbf{M}_1 = \mathbf{M}_2$.

We first show $\mathbf{Q} = \mathbf{I}$. Let $\mathbf{K}_1 := \mathrm{Cayley}(\mathrm{Skew}(\mathrm{Tril}(\mathbf{M}_1)))$ and $\mathbf{K}_2 := \mathrm{Cayley}(\mathrm{Skew}(\mathrm{Tril}(\mathbf{M}_2)))$. Notice that $\mathbf{K}_1$ and $\mathbf{K}_2$ are low-triangular due to the definition of the Cayley map. The expression $\mathbf{K}_1 = \mathbf{K}_2\mathbf{Q}$ holds only when $\mathbf{Q}$ is also a lower-triangular matrix. Because $\mathbf{Q}$ is a permutation matrix, $\mathbf{Q}$ must be an identity matrix to be lower-triangular. Thus, $\mathbf{K}_1 = \mathbf{K}_2$. Since the Cayley map is injective, we have $\mathbf{M}_1 = \mathbf{M}_2$.

## F    PROOF OF LEMMA 4

To verify this statement, we can analytically compute the Fisher-Rao metric according to its definition.

## G  Proof of Lemma 5

In a Kronecker case, we consider this spectral factorization $\mathbf{S} = \alpha[(\mathbf{B}^{(C)}\text{Diag}(\mathbf{d}^{(C)})(\mathbf{B}^{(C)})^T) \otimes (\mathbf{B}^{(K)}\text{Diag}(\mathbf{d}^{(K)})(\mathbf{B}^{(K)})^T)]$. At iteration $k$, we create a local coordinate $\boldsymbol{\eta} := (\boldsymbol{\delta}, n, \mathbf{m}^{(C)}, \mathbf{M}^{(C)}, \mathbf{m}^{(K)}, \mathbf{M}^{(K)})$ at the current point $\boldsymbol{\tau}_k := (\boldsymbol{\mu}_k, \alpha_k, \mathbf{d}_k^{(C)}, \mathbf{B}_k^{(C)}, \mathbf{d}_k^{(K)}, \mathbf{B}_k^{(K)})$ and use this local transformation map

$$
\boldsymbol{\tau}(\boldsymbol{\eta}; \boldsymbol{\tau}_k) := \begin{bmatrix} \boldsymbol{\mu}(\boldsymbol{\delta}; \boldsymbol{\tau}_k) \\ \alpha(n; \boldsymbol{\tau}_k) \\ \mathbf{d}^{(C)}(\mathbf{m}^{(C)}; \boldsymbol{\tau}_k) \\ \mathbf{B}^{(C)}(\mathbf{M}^{(C)}; \boldsymbol{\tau}_k) \\ \mathbf{d}^{(K)}(\mathbf{m}^{(K)}; \boldsymbol{\tau}_k) \\ \mathbf{B}^{(K)}(\mathbf{M}^{(K)}; \boldsymbol{\tau}_k) \end{bmatrix} = \begin{bmatrix} \boldsymbol{\mu}_k + (\mathbf{B}_k^{(C)} \otimes \mathbf{B}_k^{(K)})(\text{Diag}(\mathbf{d}_k^{(C)}) \otimes \text{Diag}(\mathbf{d}_k^{(K)}))^{-1/2}\boldsymbol{\delta} \\ \alpha_k \exp(n) \\ \mathbf{d}_k^{(C)} \odot \exp(\mathbf{m}^{(C)}) \\ \mathbf{B}_k^{(C)}\text{Cayley}(\text{Skew}(\text{Tril}(\mathbf{M}^{(C)}))) \\ \mathbf{d}_k^{(K)} \odot \exp(\mathbf{m}^{(K)}) \\ \mathbf{B}_k^{(K)}\text{Cayley}(\text{Skew}(\text{Tril}(\mathbf{M}^{(K)}))) \end{bmatrix},
$$
(35)

where $\mathbf{m}^{(C)} = [m_1^{(C)}, m_2^{(C)}, \ldots, m_{l-1}^{(C)}, -\sum_i^{l-1} m_i^{(C)}]$ has $l$ entries but only $(l-1)$ free variables since $\sum(\mathbf{m}^{(C)}) = 0$.

To verify this statement, we can analytically compute the Fisher-Rao metric according to its definition.

$$
\mathbf{F}_\eta(\boldsymbol{\delta}, n, \text{Free}(\mathbf{m}^{(C)}), \text{vecTril}(\mathbf{M}^{(C)}), \text{Free}(\mathbf{m}^{(K)}), \text{vecTril}(\mathbf{M}^{(k)}))\big|_{\boldsymbol{\eta}=\mathbf{0}}
$$
(36)

$$
= \begin{bmatrix} \mathbf{F}_{\delta\delta} & \mathbf{0} & \mathbf{0} & \mathbf{0} & \mathbf{0} & \mathbf{0} \\ \mathbf{0} & \mathbf{F}_{\alpha\alpha} & \mathbf{0} & \mathbf{0} & \mathbf{0} & \mathbf{0} \\ \mathbf{0} & \mathbf{0} & \mathbf{F}_{m^{(C)}m^{(C)}} & \mathbf{0} & \mathbf{0} & \mathbf{0} \\ \mathbf{0} & \mathbf{0} & \mathbf{0} & \mathbf{F}_{M^{(C)}M^{(C)}} & \mathbf{0} & \mathbf{0} \\ \mathbf{0} & \mathbf{0} & \mathbf{0} & \mathbf{0} & \mathbf{F}_{m^{(K)}m^{(K)}} & \mathbf{0} \\ \mathbf{0} & \mathbf{0} & \mathbf{0} & \mathbf{0} & \mathbf{0} & \mathbf{F}_{M^{(K)}M^{(K)}} \end{bmatrix}
$$
(37)

where $\text{vecTril}(\mathbf{C})$ represents the low-triangular half of $\mathbf{C}$ excluding diagonal entries and $\text{Free}(\mathbf{m})$ extracts free variables from $\mathbf{m}$.

We can see that the Fisher-Rao is block diagonal with six blocks.

The first two blocks are $\mathbf{F}_{\delta\delta} = \mathbf{I}$ and $\mathbf{F}_{\alpha\alpha} = \frac{1}{2}$. For each Kronecker factor, we have two blocks. For notation simplicity, we drop the factor index $C$ in $\mathbf{F}_{m^{(C)}m^{(C)}}$ and $\mathbf{F}_{M^{(C)}M^{(C)}}$.

For each Kronecker factor, $\mathbf{F}_{MM} = \text{Diag}(\text{vecTril}(\mathbf{W}))$, $\text{vecTril}(\mathbf{W})$ represents the low-triangular half of $\mathbf{W}$ excluding diagonal entries and its $(i,j)$-th entry is $[W]_{ij} = 4(\frac{d_i}{d_j} + \frac{d_j}{d_i} - 2) \geq 0$ and $d_i$ denotes the $i$-th entry of $\mathbf{d}_k$ for the factor. The $\mathbf{F}_{mm}$ is non-diagonal but its inverse can be computed

as $\mathbf{F}_{mm}^{-1} = 2 \begin{bmatrix} \frac{l-1}{l} & \frac{1}{l} & \cdots & \frac{1}{l} \\ \frac{1}{l} & \frac{l-1}{l} & \cdots & \frac{1}{l} \\ \vdots & \vdots & \cdots & \vdots \\ \frac{1}{l} & \frac{1}{l} & \cdots & \frac{l-1}{l} \end{bmatrix} \in \mathbb{R}^{(l-1)\times(l-1)}$ for the $(l-1)$ free variables in $\mathbf{m}$ denoted

by $\text{Free}(\mathbf{m})$. Furthermore, the natural-gradient w.r.t. $\mathbf{m}$ can also be simplified.

## H  Complete Derivation

According to Lemma 4, the Fisher-Rao metric under this local coordinate system is diagonal as

$$
\mathbf{F}_\eta(\boldsymbol{\delta}, \mathbf{m}, \text{vecTril}(\mathbf{M}))\big|_{\boldsymbol{\delta}=\mathbf{0},\mathbf{m}=\mathbf{0},\mathbf{M}=\mathbf{0}} = \begin{bmatrix} \mathbf{F}_{\delta\delta} & \mathbf{0} & \mathbf{0} \\ \mathbf{0} & \mathbf{F}_{mm} & \mathbf{0} \\ \mathbf{0} & \mathbf{0} & \mathbf{F}_{MM} \end{bmatrix}
$$
(38)

where $\mathbf{F}_{\delta\delta} = \mathbf{I}$, $\mathbf{F}_{mm} = \frac{1}{2}\mathbf{I}$, $\mathbf{F}_{MM} = \text{Diag}(\text{vecTril}(\mathbf{C}))$, $\text{vecTril}(\mathbf{C})$ represents the low-triangular half of $\mathbf{C}$ excluding diagonal entries and its $(i,j)$-th entry is $[C]_{ij} = 4(\frac{d_i}{d_j} + \frac{d_j}{d_i} - 2) \geq 0$ and $d_i$ denotes the $i$-th entry of $\mathbf{d}_k$.

Use the approximation in Eq. (9)

$$\mathbf{g}_\mu := \partial_\mu \mathcal{L} \stackrel{\text{Stein}}{=} E_{w \sim q}[\nabla_w \ell] \stackrel{\text{delta}}{\approx} \nabla_\mu \ell = \mathbf{g} \tag{39}$$

$$2\mathbf{g}_{S^{-1}} := 2\partial_{S^{-1}} \mathcal{L} \stackrel{\text{Stein}}{=} E_{w \sim q}[\nabla_w^2 \ell] - \mathbf{S} \stackrel{\text{delta}}{\approx} \nabla_\mu^2 \ell - \mathbf{S} \approx \mathcal{H} - \mathbf{S}. \tag{40}$$

where $\mathbf{g} := \nabla_\mu \ell(\boldsymbol{\mu})$ is the gradient of $\ell$, $\mathcal{H} := \mathbf{g}\mathbf{g}^T$ is a Hessian approximation.

The Euclidean gradient w.r.t local coordinate $(\boldsymbol{\delta}, \mathbf{m}, \mathbf{M})$ are

$$\mathbf{g}_\delta\big|_{\delta=0} = \mathbf{D}_k^{-1/2}\mathbf{B}_k^T\mathbf{g}_\mu \tag{41}$$

$$\mathbf{g}_m\big|_{m=0} = -\mathbf{d}_k^{-1} \odot \text{diag}(\mathbf{B}_k^T\mathbf{g}_{S^{-1}}\mathbf{B}_k) \tag{42}$$

$$\mathbf{g}_{\text{vecTril}(M)}\big|_{M=0} = 4\text{vecTril}(\mathbf{B}_k^T\mathbf{g}_{S^{-1}}\mathbf{B}_k\mathbf{D}_k^{-1} - \mathbf{D}_k^{-1}\mathbf{B}_k^T\mathbf{g}_{S^{-1}}\mathbf{B}_k) \tag{43}$$

where $\mathbf{D}_k := \text{Diag}(\mathbf{d}_k)$. Recall that we use a gradient outer product $\mathcal{H} = \mathbf{g}\mathbf{g}^T$ as a Hessian approximation in $2\mathbf{g}_\Sigma$, where $2\mathbf{g}_{S^{-1}} \approx \mathbf{g}\mathbf{g}^T - \mathbf{S} + \lambda\mathbf{I}$ considered in RMSProp, where $\lambda\mathbf{I}$ is included for damping.

The FIM can still be singular when $\mathbf{d}$ has repeated entries (i.e., $d_i = d_j$ for $i \neq j$) since $\mathbf{F}_{MM}$ can be singular. We can use the Moore-Penrose inverse when computing the inverse. Thanks to this coordinate system, $\mathbf{F}_{MM}$ is indeed a diagonal matrix.

Thus, we can simplify the RGD update as

$$\begin{bmatrix} \boldsymbol{\delta} \\ \mathbf{m} \\ \text{vecTril}(\mathbf{M}) \end{bmatrix} \leftarrow \begin{bmatrix} \mathbf{0} - \beta_1\mathbf{F}_{\mu\mu}^{-1}\mathbf{g}_\delta \\ \mathbf{0} - \beta_2\mathbf{F}_{mm}^{-1}\mathbf{g}_m\big|_{m=0} \\ \mathbf{0} - \beta_2\mathbf{F}_{MM}^{-1}\mathbf{g}_{\text{vecTril}(M)}\big|_{M=0} \end{bmatrix} = \begin{bmatrix} \mathbf{0} - \beta_1\text{Diag}(\mathbf{d})^{-1/2}\mathbf{B}^T\mathbf{g}_\mu \\ \mathbf{0} - 2\beta_2\mathbf{g}_m\big|_{m=0} \\ \mathbf{0} - \beta_2\mathbf{F}_{MM}^{-1}\mathbf{g}_{\text{vecTril}(M)}\big|_{M=0} \end{bmatrix}, \tag{44}$$

where we introduce another learning rate $\beta_2$ when updating $\mathbf{d}$ and $\mathbf{B}$.

Note that when $d_i \neq d_j$ for $i \neq j$, the $(i,j)$-th entry of the natural gradient w.r.t. $\mathbf{M}$ is

$$[\mathbf{F}_{MM}^{-1}\mathbf{g}_{\text{vecTril}(M)}\big|_{M=0}]_{ij} = (\mathbf{B}_k^T\mathbf{g}_{S^{-1}}\mathbf{B}_k)_{ij}/(d_i - d_j). \tag{45}$$

When $d_i = d_j$, we simply set the corresponding entry to be zero due to the Moore-Penrose inverse.

Finally, we can re-express the above update as:

$$\begin{bmatrix} \boldsymbol{\mu}_{k+1} \\ \mathbf{d}_{k+1} \\ \mathbf{B}_{k+1} \end{bmatrix} \leftarrow \begin{bmatrix} \boldsymbol{\mu}_k - \beta_1\mathbf{B}_k\text{Diag}(\mathbf{d}_k)^{-1}\mathbf{B}_k^T\mathbf{g}_\mu \\ \mathbf{d}_k \odot \exp[0 + 2\beta_2\mathbf{d}_k^{-1} \odot \text{diag}(\mathbf{B}_k^T\mathbf{g}_{S^{-1}}\mathbf{B}_k)] \\ \mathbf{B}_k\text{Cayley}(\text{Tril}(\mathbf{U}) - [\text{Tril}(\mathbf{U})]^T) \end{bmatrix} \tag{46}$$

where the $(i,j)$ entry of $\mathbf{U}$ is $U_{ij} = 0 - \beta_2(\mathbf{B}_k^T\mathbf{g}_{S^{-1}}\mathbf{B}_k)_{ij}/(d_i - d_j)$ for $i \neq j$ and $U_{ij} = 0$ when $d_i = d_j$ thanks to the Moore-Penrose inverse.

