# OpenReview forum: "Fast Fractional Natural Gradient Descent using Learnable Spectral Factorizations"
_ICLR.cc/2025/Conference — Submitted to ICLR 2025_

### Official Review · Reviewer_YraM · 2024-11-04

**Soundness:** 3
**Presentation:** 2
**Contribution:** 2
**Rating:** 5
**Confidence:** 3

**Summary:**

The paper introduces a new update rule for the approximation of Fisher Information matrix in the natural gradient descent and its variants like RMSprop and Adam(W). The intuition is based on the observation that the preconditioner can be interpreted as the inverse covariance matrix of a Gaussian variational distribution (Lin et al. 2024). The authors propose to learn a decomposition of the preconditioner so that the fractional inversion can be computed efficiently. Numerical experiments show that the iterates generated by such a method have comparable efficiency in approximating the Fisher information and good performance when used to train neural networks.

**Strengths:**

1. The paper addresses a relevant and interesting problem in optimizing deep neural networks, which may inspire future research ideas.

2. In addition to introducing the counterpart for the classical NGD methods, the paper also introduces the update rule that works for methods like Shampoo, which preconditions on both sides of the matrix iterate.

3. The paper clearly describes its own contribution and credits the contribution of the previous works.

**Weaknesses:**

1. The main motivation of the method, which is the flexibility of choosing p, is mentioned but not very well established. I think the authors should at least make an effort on the numerical side. They mention that "This shows that the potential of using other fractional roots." by the end of the section 4, which seems weak as this is the main reason for potential practitioners to use this method over its simpler counterpart. Some results showing that the best training result can only be obtained by some p other than 1 or 2 can be very persuasive in this case.

2. In terms of the computational cost, the comparison between this method and previous methods is not clear. The authors calibrated the running time for each method in section 4 by adjusting the number of iterations to update the preconditioner, but it is still vaguely defined and not quantified. One good comparison is to those methods which directly compute the matrix decomposition using high-precision arithmetic. I think this will establish the other important motivation of the method.

3. The method suffers from a per-iteration error of $O(\beta_2^2)$, which is non-negligible for constant $\beta_2$. Also, the inverse approximation when calculating the Cayley map introduces another source of inaccuracy.

4. There are some notation ambiguity in the paper. For example, Kronecker product and functions like Mat($\cdot$) are not defined clearly. Diag($\cdot$) and diag($\cdot$) are used interchangeably. Eq 3 seems to come from nowhere when it first appears in the paper (no introduction before and after, no definition of $\mathcal{H}$). Claim 2 eventually becomes Lemma 2 in the appendix.

**Questions:**

My questions mostly overlap with the confusion I have in the Weaknesses section above. It would be great if authors could address those concerns, in which case I would happily increase my score.

---

### Official Review · Reviewer_CVs4 · 2024-11-04

**Soundness:** 4
**Presentation:** 3
**Contribution:** 4
**Rating:** 8
**Confidence:** 4

**Summary:**

This paper proposes a stable and efficient Riemannian framework to learn eigen-factorized Fisher estimations, allowing for the cheap application of any fractional powers. The efficacy of this approach is demonstrated in numerical tasks including positive-definite matrix optimization and low-precision NN training.

**Strengths:**

1. This approach allows for the cheap application of arbitrary fractional powers, which distinguish it from other similar RGD approaches (e.g. Cholesky-based RGD).
2. This approach circumvents the numerical instabilities of matrix decompositions, making it easy to implement and more practical.
3. This paper is an extension based on previous literatures which viewed learning the curvature approximation (preconditioner) as learning the covariance of a Gaussian variational distribution by performing RGD on the manifold of dense or Kronecker-factorized positive-definite matrices. This paper further separates the diagonal part and orthogonal part by spectral decomposition, and incorporates the new constraints arising from the spectral decomposition for the Fisher-Rao metric in derivation.

**Weaknesses:**

1. No numerical performance in large-scale setting is presented.

**Questions:**

1. How did you generate matrix $Q$ in the positive-definite matrix optimization problem of Section 4?
2. For the low-precision NN training in Section 4, you claimed that $p=1$ is better than $p=2$, but the difference is not significant from the plots presented. Do you have any other numerical result showing a more significant discrepancy?
3. Did you have a complete derivation of the procedure for Kronecker-structured Spectral Parametrizations?

---

> ### Comment · Reviewer_CVs4 · 2024-11-26
> **Add**
>
> The benefit of arbitrary fractional power needs to be further explained.

---

> > ### Author Response · Authors · 2024-11-27
> > **Thanks for the additional comment**
> >
> > We thank the reviewer for the additional comment and the appreciation of the Riemannian idea. We will discuss this with more empirical results in the next revision.

---

### Official Review · Reviewer_Db9H · 2024-11-04

**Soundness:** 3
**Presentation:** 1
**Contribution:** 2
**Rating:** 3
**Confidence:** 2

**Summary:**

This paper proposes an adaptive update scheme for full-matrix spectral factorization. This method serves as the efficient approximation of the preconditioning matrix, thus, allowing to perform a step of natural gradient descent faster.

**Strengths:**

**S1:** The authors study a new form of the preconditioneer – spectral factorization of the curvature approximation. They show, that for such a preconditioneers can b eapplied an update rule allowing to learn the spectral factorization at each step of the optimization problem.

**S2:** The approach of the paper extends to the Kronecker factorization and analysis of the spectral decomposition for the Fisher-Rao metric.

**Weaknesses:**

**W1:** The writing of the paper is hard to follow.

**W2:** The are no explicit formulation of lemmas in the paper.

**W3:** The key points in the theoretical analysis is missing.

**Questions:**

**Q1:** In Line 318, the authors manage a phrase: ''When changing coordinates, the metric representation has to be changed accordingly Lee, (2018) to make RGD invariant to coordinate transformation''. Please, could the authors clarify a steps like that with more formal presentation?

**Q2:** Can the authors clarify a conceptual difference between the *Procedure for Full-matrix Spectral Parametrizations* and procedure in *Riemannian Approach for Obtaining Root-free Update Schemes*? As, naturally, the *Step 1* and *Step 2* of the latter approach describes the whole pipeline in general.

**Q3:** Why does the *Claim 5* not considered as a definition?

**Q4:** Could the authors clarify, please, how does it follows from Figure 4, that fraction roots are better than p=2?

**Possible suggestions:**

1) Rename ''default update scheme'' and ''our scheme'' with more sophisticated names.
2) Explicitly write Lemmas 1-5.
3) Formally define Fisher-Rao metric in the gentle introduction to the approach.



John M Lee. Introduction to Riemannian manifolds, volume 2. Springer, 2018.

---

### Official Review · Reviewer_impa · 2024-11-06

**Soundness:** 2
**Presentation:** 2
**Contribution:** 3
**Rating:** 5
**Confidence:** 2

**Summary:**

In the paper, the author propose a novel optimization method for deep learning. It aims to learn spectral factorization of Fisher preconditioning without matrix decomposition. By knowing the spectral factorization it is possible to apply arbitrary power of the preconditioning. Finally, the authors provide evidence of practical effectiveness and performance of the proposed method.

**Strengths:**

The proposed method is focused on interesting and valuable problems to solve. The paper has a good and easy-to-follow introduction which motivates the proposed method. The method has a big potential from the practical perspective as it exploits curvature information.

**Weaknesses:**

1. The style of presentation is confusing in some parts of the paper. Next, I list some examples: a) Figure 1 shouldn’t be presented as is. It has two methods inside and half of the paper description. It ruins the style and flow of the paper. b) The authors present theoretical results as “claims”, however, next they prove it as lemmas. So, it would be reasonable to formulate them as lemmas in the main text as well. c) In lines 151-161 and 242-248, there is an extra spacing which shouldn’t be there.  d) Some colorings are confusing. For example, line 353, 745, 833, 932. It might be forgotten highlights of the changes.

2. The experimental section. As the paper doesn’t provide convergence proofs, we can say it is mostly an empirical paper backed by theoretical intuition. a) Hence, the extensive experimental results are expected. In the paper, only one figure presents the performance of the methods for 3 vision transformers on one dataset. It does not seem enough for an experimental paper. So, I would recommend adding experiments for common DL benchmarks to understand the methods' performance in comparison to state-of-the-art methods. Some examples can be found here https://mlcommons.org/benchmarks/algorithms/ b) Another issue is that the experiments’ description is lacking proper details and descriptions. It is limiting the reproducibility of the results. There are no methods parameters, learning rates, and so on. The attached code may help as well.

Some minor misprints: a) Please, unify the use of “RMSprop”, as currently there are multiple variants of it (RMSprop, RmsProp, RMRprop) b) Line 709: misprint in “Given”

**Questions:**

1) Is the method implemented as a PyTorch optimizer?

2) How many parameters are required for the method's performance?

---

### Official Review · Reviewer_rMhK · 2024-11-07

**Soundness:** 3
**Presentation:** 2
**Contribution:** 3
**Rating:** 5
**Confidence:** 4

**Summary:**

The paper proposes a novel approach to fractional natural gradient descent (FNGD) by a learning method, which addresses the challenges of computing the matrix fractional powers. The authors present a Riemannian framework to learn eigen-factorized Fisher estimations on-the-fly, which allows the efficient application of arbitrary fractional powers without requiring matrix decompositions. Numerical experiments show the effectiveness in positive-definite matrices optimization problems and neural networks training tasks.

**Strengths:**

The introduction of learnable spectral factorization to FNGD is an innovation that addresses computational bottlenecks in second-order methods, such as those found in Shampoo or KFAC.

The authors effectively address numerical instability, making the proposed method suitable for low-precision arithmetic, which may improve the efficiency in large-scale machine learning tasks.

**Weaknesses:**

To strengthen the argument for using this method over existing alternatives, an ablation study examining the effects of different spectral factorization techniques on convergence speed and generalization performance would be beneficial.

While the paper demonstrates scalability in terms of Kronecker-structured preconditioners, additional details on computational complexity and memory requirements—particularly in comparison to popular methods like Adam or Shampoo—would offer a clearer understanding of the associated trade-offs.

**Questions:**

Could you provide a more intuitive explanation of the advantages of using spectral parametrization over traditional matrix decomposition methods, especially regarding computational efficiency?

It would also be valuable to include more comparisons with recent advancements in adaptive gradient methods, particularly as applied to large-scale language models.

Additionally, a clearer summary of the implications of using arbitrary fractional powers in gradient descent in both the introduction and conclusion would enhance accessibility, helping readers better grasp the broader impact of this approach.

Why was the KFAC method excluded from the numerical comparison?

As an early suggestion, moving the Kronecker extension to the appendix could allow more space for a structured development of the core method in the main text, which may improve readability.

**Details Of Ethics Concerns:**

N.A.

---

### Official Review · Reviewer_qdL2 · 2024-11-08

**Soundness:** 2
**Presentation:** 1
**Contribution:** 1
**Rating:** 3
**Confidence:** 3

**Summary:**

The paper proposes a method for fractional natural gradient descent (FNGD) using learnable spectral factorizations. It introduces a way to factorize the Fisher matrix and apply fractional powers, aimed at enhancing neural network training.

**Strengths:**

Consider a unified framework for adaptive methods through the fractional power of the Fisher matrix.

**Weaknesses:**

Clarity Issues: Notation and concepts are unclear, making the method difficult to follow. For example:
- What is $B$ matrix?
- The purpose of learning the Kronecker factorization is under-explained
- What is matrix H? (In line 101)
- Meaning of the “mat” notation (line 107)
- What are matrices $S^K$ and $S^C$?
- What is $\otimes$ product?
- Description of the “Gaussian problem”?
For that concept or notations, the authors should give an appropriate introduction and explanation and references

Weak Motivation:
The paper provides an insufficient explanation as to why this complex approach is necessary or how it significantly advances existing methods. It does not provide new insight or guidelines (or at least does not explain clearly) that can potentially be utilized to result in efficient optimization or training.

Ambiguity in Implementation:
The handling of factorization ambiguities in high-dimensional settings is unclear, particularly in terms of computational impact and scalability.

**Questions:**

NA

---

### Official Review · Reviewer_x133 · 2024-11-08

**Soundness:** 2
**Presentation:** 1
**Contribution:** 2
**Rating:** 5
**Confidence:** 2

**Summary:**

This paper introduce a unifying framework way to calculate Fractional natural gradient descent estimates such as RMSprops or Shampoo by considering a diagonalization of the curvature matrix $S$ as $B diag(d) B^\top$ for orthogonal matrix $B$, and updating $B$ and $d$ instead. This parameterization allows them to calculate the fractional power easily as $B \diag(d^{-1/p})B^\top$. To update the parameters $B$ and $d$, the challenge is that there are constraints on the space of the middle matrix that has to be diagonal, and $B$ has to be orthogonal. The authors then seems to use a reparameterization of the space  $(B,d)$ in what they call a "root-free" approach, so that they can take the Riemannian gradient steps in that reparameterization with any constriants, in a specific manifold metric that is related to a concept called "Gaussian variational distrbution."  To do so, the they use previous ideas going back to work of Amari et al.

**Strengths:**

The authors introduce a unifying framework for a class of preconditioned gradient agorithms as special cases of fractional natural gradient descent, and gives a general framework to implement it more efficiently in a spectral parameterization that is motivated by a Riemannian optimization algorithm in a reparameterization of the space to the normal coordinates.

**Weaknesses:**

1. It is nots clear what is their contribution and what has been don ein previous work.
2. Description of the use of the gaussian variational approach in line 286 is very vague, it is not even clear why this approach is well-motivated or what is $\ell$.
3. Authors keep taking about suing local coordinates but it's not exactly what coordinate system for the manifold of matrices they are referring to, do you mean the normal coordinate system using the xponential map?

4. The definition of function $Tril$ is unclear, and authors define what they mean by $Cayley$ in the middle of the paper.
5. The high level ideas are vague and include terms that is not clear what they exactly refer to.
6. It would be nice if authors can provide further motivation or example on which practical case the kronecker-structured spectral parameterization is useful.
7. It is not clear what is the manifold metric the authors consider in their Riemannian appraoch. they connection of the gaussian variational framework, which seems to go back to work of Amari,  to obtaining a manifold metric seems unclear.

Other issues:
H is not defined in equation 3.
U and Tril are not defined in algorithm box on line 162
Line 107 the notation is super unclear, is Mat(g) the diagonal matrix with entries of g? If so what is the difference between $G^\top G$ and $G G^\top$, and $S_C, S_K$?
Authors refer to “root-free” method but they don’t define what they mean
Equation 10 is not clear at all
Authors don't provide any intuition of their approach/contribution e.g. for equation 10 or for the algorithm box in line 162

Minor issue:
Claim 1 is named Lemma 1 in appendix

**Questions:**

1. what is the meaning of "root-free" scheme?
2. Can you please elaborate on how you define the sequence of the curvature matrices in "Iterate matching" scheme to measure the performance of your method?
3. Can authors reference which work shows superior generalization of the FNGD for p's either than 2, 4, 1?

---

### Author Response · Authors · 2024-11-27
**Thanks for your reviews**

We thank the reviewers for taking time to review our work.
We will use the feedback to improve our work and incorporate it into the next revision of the paper.

---

### Meta-Review · Area_Chair_BEMz · 2024-12-18

**Metareview:**

This work unifies popular optimization methods through Fractional Natural Gradient Descent (FNGD), which preconditions gradients using fractional powers of the Fisher inverse. To address limitations in existing methods that rely on expensive matrix computations, the authors propose a Riemannian framework that learns eigen-factorized Fisher estimations on the fly, enabling efficient application of arbitrary fractional powers without requiring matrix decompositions. This approach is stable in half precision and demonstrates effectiveness in optimizing positive-definite matrices and training neural networks.

Most reviewers indicate that paper offers an interesting unifying framework for several popular optimization methods. However, the main concern raised by several reviewers was the lack of clarity in the presentation, the contributions of the paper, and the ambiguity in the implementation of the method.

The AC shares these concerns and believes the paper would benefit from a major revision to address the highlighted issues.

**Additional Comments On Reviewer Discussion:**

Seven reviews were collected for this paper, with six recommending rejection and one recommending acceptance. The AC agrees with the majority vote, supporting a rejection due to the critiques raised by the reviewers.

---

### Decision · Program_Chairs · 2025-01-22

Reject